# Pushing on Multilingual Reasoning Models with Language-Mixed Chain-of-Thought

**Guijin Son**[1,6]     **Donghun Yang**[2]     **Hitesh Laxmichand Patel**[3]     **Amit Agarwal**[3]
**Hyunwoo Ko**[1]     **Chanuk Lim**[2]     **Srikant Panda**[3]     **Minhyuk Kim**[4]     **Nikunj Drolia**[7]
**Dasol Choi**[5]     **Kyong-Ha Lee**[2*]     **Youngjae Yu**[6*]

[1]OneLineAI     [2]KISTI     [3]Oracle AI     [4]Korea University     [5]AIM Intelligence
[6]Seoul National University     [7]University College Dublin     [8]Yonsei University

spthsrbwls123@yonsei.ac.kr

## Abstract

Recent frontier models employ long-chain-of-thought reasoning to explore solution spaces in context and achieve stronger performance. While many works study distillation to build smaller yet capable models, most focus on English and little is known about language-specific reasoning. To bridge this gap, we first introduce **Language-Mixed CoT**, a reasoning schema that switches between English and a target language, using English as an anchor to excel in reasoning while minimizing translation artifacts. As a Korean case study, we curate **Yi-Sang**: 5.79M native-Korean prompts from web Q&A, exams, STEM, and code; 3.7M long reasoning traces generated from Qwen3-32B; and a targeted 260k high-yield subset. We train nine models (4B–35B) across six families (Qwen2.5, Llama-3.1, Gemma-3, etc). Our best model, **KO-REAson-35B**, achieves state-of-the-art performance, with the highest overall average score ($64.0_{\pm2.5}$), ranking first on 5/9 benchmarks and second on the remainder. Smaller and mid-sized models also benefit substantially, with an average improvement of $+18.6$ points across the evaluated nine benchmarks. Ablations show **Language-Mixed CoT** is more effective than monolingual CoT, also resulting in cross-lingual and multi-modal performance gains. We release our data-curation pipeline, evaluation system, datasets, and models to advance research on language-specific reasoning.[1]

## 1 Introduction

Test-time scaling amplifies reasoning by allocating more samples or steps, enabling exploration, and self-correction (Jones, 2021). Recent advances show that large language models can internalize similar exploratory behavior (Gandhi et al., 2025) through longer chain-of-thought (CoT) acquired during training. Specifically, such behaviors stem during the post-training phase through reinforcement learning with verifiable rewards (RLVR) (OLMo et al., 2024; Guo et al., 2025). Unfortunately, such methodologies tend to be effective only for strong base models with large parameters (Yang et al., 2025; Rastogi et al., 2025). Therefore, open efforts have centered on distillation from frontier teacher models, combining systematic prompt collection with response generation and quality filtering (Muennighoff et al., 2025; Bercovich et al., 2025; Guha et al., 2025; Hugging Face, 2025). Such pipelines, however, overwhelmingly target English and, to a lesser extent, Chinese (Liu et al., 2025a), leaving open how to achieve language-specific reasoning. To bridge this gap, we study how to construct a reasoning model for a mid-resource language through a focused case study in Korean.

We start from the empirical observation that pipelines relying heavily on translated corpora (lightblue, 2025; Lee et al., 2025a) exhibit degraded response quality from translation artifacts (Park et al., 2025; Li et al., 2025a) and poor robustness to everyday, colloquial expressions that rarely appear

---

[*]Corresponding Authors

[1]Data and Model Collection:https://huggingface.co/KOREAson

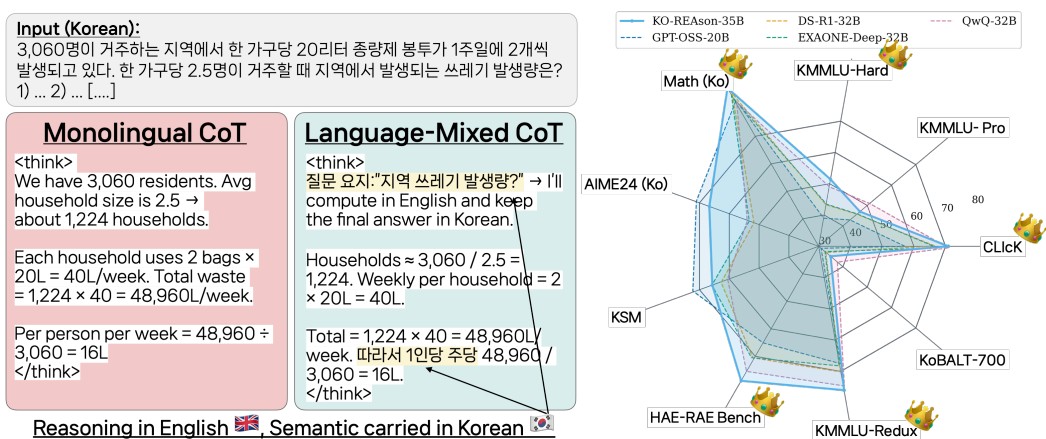

Figure 1: **(Left) Thinking styles.** Red: monolingual CoT carried out entirely in English. Blue: our proposed **Language-Mixed CoT**, which alternates between English (anchor) and Korean (target). **(Right) Performance comparison of KO-REAson-35B (ours, solid line) with DeepSeek-R1-32B, Exaone-Deep-32B, GPT-OSS-20B, and QwQ-32B.** KO-REAson-35B achieves top-tier performance, ranking first or second on all tasks.

in translated text. To address this, we propose a two-step approach: *(i) data curation*, where we collect 5.79M Korean, user-authored Q&A prompts from the web to ensure broad coverage of natural, in-the-wild language; and *(ii) reasoning supervision*, where, when generating long reasoning traces with Qwen3-32B (Yang et al., 2025), we enforce **Language-Mixed CoT**, which allows the model to switch freely during the *Think* step between an anchor language (English) and the target language (Korean). This enables the model to leverage the anchor language's reasoning capabilities while preserving the semantics of the target language. In our experiments, Language-Mixed CoT consistently outperforms monolingual CoT, with larger gains on reasoning-heavy tasks relative to Korean-only, and on cultural understanding-heavy tasks relative to English-only.

The collected dataset, YI-SANG, comprises 5.79M prompts paired with 3.7M long reasoning traces. To the best of our knowledge, this is the largest publicly documented post-training resource for the Korean language. To chart an affordable path to strong reasoning models, we conduct over 100 ablations (some scaling to thousands of H100 GPU-hours) covering teacher models, augmentation schemes, and seed sources, and we iteratively filter patterns that produce loss spikes. This process yields a downsampled YI-SANG-HQ of 260k high-yield examples, on which we train the KO-REAson series. As shown in Table 4, **KO-REAson-35B** outperforms state-of-the-art models trained on closed data (GPT-OSS-20B (Agarwal et al., 2025b), R1-Distill-32B (Guo et al., 2025), QwQ-32B (Team, 2025), EXAONE-Deep-32B (Research et al., 2025)) on average across nine tasks. We further demonstrate that these gains are consistent across model families and scales by training nine models (4B–35B) spanning six families. Finally, we observe cross-lingual and multi-modal gains, despite training only on Korean text. Taken together, these results indicate that careful prompting and large-scale data collection can build open-recipes to rival closed systems.

Our contributions are summarized as follows:

- We introduce YI-SANG, the largest publicly documented post-training dataset for Korean to date, comprising 5.79M prompts and 3.7M long reasoning traces, plus a 260k high-yield subset (**YI-SANG-HQ**) distilled via extensive ablations.

- We propose **Language-Mixed CoT**, a supervision scheme that lets models switch between an anchor language (English) and the target language (Korean) during the *Think* step, yielding significant gains over monolingual CoT baselines.

- We train and release the **KO-REAson** series (4B–35B across five families) under the Apache-2.0 license, surpassing closed systems of comparable scale on nine benchmarks.

## 2 PRELIMINARIES AND RELATED WORKS

Recent work has pushed long reasoning into the mainstream. o1 (Jaech et al., 2024) showed that extending the 'thinking length' of a model improves performance, while R1 (Guo et al., 2025) revealed how long reasoning traces are structured and how to build models capable of such capability. DeepSeek also demonstrated that SFT-distilled (e.g., DeepSeek-Distill-R1) variants can inherit much of this ability from supervised fine-tuning alone. Subsequent efforts

Table 1: **Performance of Qwen2.5-1.5B-Instruct before and after fine-tuning.** Fine-tuning on translated OpenThoughts-114K for five epochs improves performance on MATH while degrading on *HAE-RAE Bench*.

| Model | HRB | MATH |
|---|---|---|
| Qwen2.5-1.5B | **35.24** | 25.48 |
| + TRANSLATED OT | 15.34 | **74.35** |

span online RL (Yu et al., 2025; Chen et al., 2025; Luo et al., 2025), offline RL (Research et al., 2025; Wen et al., 2025), and pure SFT (Muennighoff et al., 2025; Guha et al., 2025). A consistent pattern emerges: successful online RL from a cold start typically requires (i) a strong base model (often $\geq$30B, with solid math/coding priors) (Yang et al., 2025; Rastogi et al., 2025), (ii) a reliable process or reward model (Liu et al., 2025c; He et al., 2025), and (iii) large-scale, high-quality data (e.g., Numina-Math (LI et al., 2024)). These requirements increase cost and brittleness, concentrating progress in high-resource languages such as English and Chinese.

Much less is known about bootstrapping reasoning models in mid-resource languages. Directly replicating high-resource pipelines is often infeasible due to weaker base models and limited high-quality data. Prior works focus on leveraging carefully designed SFT mixtures or learning objectives to bring non-English representations closer to English (Zhu et al., 2024; **?**; Agarwal et al., 2025a; Lai & Nissim, 2024; Chen et al., 2024) or explore cross-lingual transfer either by English training (Yong et al., 2025; Ranaldi & Pucci, 2025) or small-scale translated datasets (Son et al., 2025; Pipatanakul et al., 2025). Following such, we first train Qwen2.5-1.5B-Instruct on translated data from OpenThought1 (Guha et al., 2025). As shown in Table 1, this model achieves improved performance on MATH but suffers a substantial drop on HAE-RAE Bench (HRB) (Son et al., 2023), a Korean culture benchmark. This gap motivates us to develop a reliable and practical recipe for building a reasoning model that attains robust performance across diverse domains, rather than focusing only on mathematical reasoning.

Our work differs from previous works by going beyond translation. We collect native prompts, systematically curate for quality, and introduce **Language-Mixed CoT** as a more effective supervision signal. By varying only supervision format (long vs. short; language-mixed vs. monolingual), we isolate supervision effects from optimization confounds and provide mid-size models a stable path to long-reasoning behavior without RL. We validate this methodology in Korean, an apt testbed: a mid-resource language with an active LLM research ecosystem, scratch-trained base models (Bak et al., 2025; Lab, 2025; KISTI, 2024), dedicated general-knowledge (Son et al., 2024; Hong et al.,

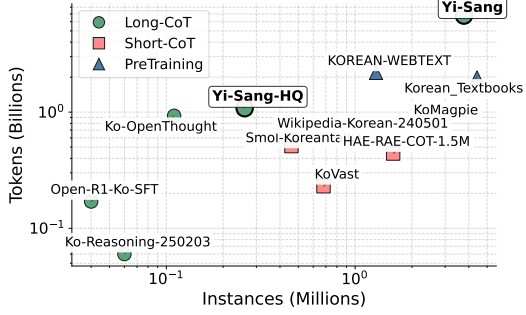

Figure 2: **An overview of publicly available Korean datasets.** YI-SANG is larger than any fine-tuning dataset or pretraining corpus, with 6.77B tokens.

2025) and reasoning benchmarks (Ko et al., 2025), and sufficiently large web corpora for data construction. The proposed dataset, YI-SANG, is not only the largest Korean post-training corpus (Figure 2), but also a methodological contribution: a pipeline for converting noisy internet prompts into high-quality supervision. Our empirical results demonstrate its effectiveness and offer a reproducible path for mid-resource communities to build competitive reasoning models.

# 3 EXPERIMENTAL SETUP

## 3.1 TRAINING DETAILS

**Models**  To ensure robustness in our ablations, we run experiments on two base models: *Gemma-3-4B* (Team et al., 2025) and *Kanana-1.5-8B* (Bak et al., 2025). After determining the high-yield subset, we evaluate its efficacy by training across a broader set of models, including *Gemma-3-4B/12B, A.X-3.1-7B/35B* (Lab, 2025), *Kanana-1.5-8B, Llama-3.1-8B* (Grattafiori et al., 2024), *KONI-Llama-*

*3.1-8B* (KISTI, 2024), and *Qwen2.5-7B/14B* (Qwen et al., 2025). All experiments are conducted with the instruction-tuned versions of the models. See Appendix B.1 for further details on each model.

**Training Settings** All training runs use a minimum of 50,000 data points unless otherwise specified. Each experiment (including ablations) is trained for five epochs, except for *A.X-3.1-35B*, which we train for three epochs due to computational constraints. For further details on the hyperparameters used throughout training, see Appendix B.

## 3.2 EVALUATION DETAILS

**Benchmarks** In this work, we divide our evaluation suite into two parts: a held-in set, used for routine monitoring during training and ablation studies, and a held-out set, evaluated once after all ablations and final training are complete. This is to support rapid iteration and prevent inadvertent overfitting to benchmarks during iterative training ablations.

- **Held-in** consists of four benchmarks. *MCLM* (Son et al., 2025) is a translated collection of math problems from MATH500 and AIME2024, originally drawn from Olympiads, designed to test deep chain-of-thought reasoning rather than surface recall. *KMMLU-Redux* (Hong et al., 2025) is a quality-controlled, down-sampled version of KMMLU (Son et al., 2024) that maintains correlations with the full suite while reducing evaluation cost; importantly, it spans both factual knowledge (e.g., history, law, medicine) and reasoning-intensive domains (e.g., mathematics, engineering, science). *HAE-RAE Bench* (Son et al., 2023) assesses Korean linguistic and cultural competence, covering vocabulary, reading comprehension, and historical content. For medical ablations, we also include *ClinicalQA*, a Korean clinical QA benchmark derived from medical licensing examinations, consisting of problems based on chief complaints and medical specialties.

- **Held-out** covers a broader set of benchmarks used only after all training ablation is done. *KMMLU-Hard* (Son et al., 2024) is adversarially filtered version of KMMLU for highest difficulty. *KMMLU-Pro* (Hong et al., 2025) contains expert-level professional licensure questions across 14 different categories, including Medicine, Finance, and Law. *KSM* (Ko et al., 2025) is a set of competition-style mathematics problems from Korean contests. *CLIcK* (Kim et al., 2024) aggregates factual questions from Korean exams and textbooks across 11 categories, providing a measure of Korean general world knowledge. Finally, *KoBALT-700* (Shin et al., 2025) is a linguistics-focused benchmark of 700 expert-written items that span syntax, semantics, morphology, phonology, and pragmatics, used to test fine-grained Korean linguistic competence.

**Evaluation Setup** All evaluations are run with vLLM (Kwon et al., 2023) under the following configuration: `temperature`=0.7, `top_p`=0.9, and `max_tokens`=32,768. Models are instructed to present the final answer wrapped in `\boxed{...}`, and we use math-verify [2] to validate the boxed value; outputs without a valid answer are marked incorrect. All ablations use a single evaluation; for the main experiments, we run three independent trials and report mean $\pm$ standard error.[3]

## 4 LANGUAGE-MIXED CHAIN-OF-THOUGHT

When constructing multilingual reasoning data in a target language (other than English), a central question is how to represent the reasoning process: *should it be written in the target language or left in English?* Prior work has typically chosen one of two monolingual setups, either entirely in English (Pipatanakul et al., 2025; Ha, 2025; Son et al., 2025) or entirely in the target language (lightblue, 2025; Lee et al., 2025a). Our initial exploration reveals critical shortcomings in both. Reasoning in English on Korean prompts introduces translation noise: prompts are often mistranslated, especially in culture-specific contexts, and over time, errors accumulate, leading the model to drift off topic once it "forgets" the original Korean wording. Conversely, reasoning in Korean produces notable drops in reasoning capability (Ko et al., 2025), and extended training in Korean induces distributional drift in English-pretrained bases (Hong et al., 2024), degrading their original strengths.

---

[2] `https://github.com/huggingface/Math-Verify`
[3] See Section C.2 for more details.

Table 2: **Language-Mixed CoT (ours) outperforms monolingual CoTs across models and sizes.** Compared with English- or Korean-only CoT, Language-Mixed CoT yields higher scores for both Gemma (4B) and Kanana (8B). Highest scores per column are highlighted in green. Abbreviations: HRB = HAE-RAE Bench; KMMLU-R = KMMLU-Redux.

| CoT Lang. | Gemma-3-4B | | | Kanana-1.5-8B | | |
|---|---|---|---|---|---|---|
| | HRB | MCLM | KMMLU-R | HRB | MCLM | KMMLU-R |
| English | 50.3 | 48.1 | 52.2 | 66.2 | **60.5** | 64.0 |
| Korean | 40.6 | 25.6 | 42.5 | 67.2 | 31.8 | 53.4 |
| Language-Mixed$_{ru/ko}$ | 46.7 | 22.5 | 44.1 | 67.6 | 28.7 | 50.4 |
| Language-Mixed$_{zh/ko}$ | 48.2 | 26.3 | 45.3 | 68.8 | 25.6 | 51.1 |
| Language-Mixed$_{en/ko}$ | **54.9** | **55.8** | **53.0** | **74.6** | 57.4 | **64.4** |

To address both issues, we propose **Language-Mixed CoT**[4] (See Figure 1 for example). During the Think phase, the model code-switches, performing most logical scaffolding in English while preserving key Korean terms and quotations. This keeps faithfulness to the prompt without sacrificing reasoning power. To generate **Language-Mixed CoT**, we prompt the teacher to preserve named entities, quoted spans, and key terms in Korean while generating the rest of the reasoning in English. After generation, we apply a regex-based filter to discard samples whose Korean-character ratio lies outside 5% and 20%. In Table 2, we train five variants: an English and Korean-only model, and three language-mixed models that combine Korean with English, Chinese, or Russian. The choice of Chinese and Russian follows Qi et al. (2025): Chinese is culturally and historically closer to Korean, whereas Russian is relatively distant. Notably, language-mixed CoT with English anchoring outperforms other settings in most cases. Interestingly, Gemma3-4B shows gains on HRB and KMMLU-R even with Russian- or Chinese-anchored CoT, whereas Kanana-1.5 does not. We suspect this difference is driven by the pretraining mixtures: Gemma3-4B is a multilingual model that includes substantial Russian and Chinese data, while Kanana-1.5-8B is pretrained only on English and Korean. *Most importantly, however, improvements on MCLM (math) emerge only when using English-anchored CoT.*

## 5 YI-SANG INSTRUCT

Despite many efforts to distill frontier models into smaller open models, only a few manage to collect their own training corpus; most reuse or repackage existing datasets (Ye et al., 2025; Guan et al., 2025; Hugging Face, 2025; Hu et al., 2025). This pattern is also common in multilingual settings and materially affects outcomes: models trained through such pipelines lack robustness to everyday colloquial expressions that rarely appear in translated text. To pursue a more robust multilingual reasoning, we decided to construct our own dataset. This section describes our instruction collection (Section 5.1), response generation process (Section 5.2) for building **YI-SANG**, and presents ablations used to derive the high-yield subset **YI-SANG-HQ** (Section 5.3).

### 5.1 INSTRUCTION COLLECTION

**Seed Instruction Collection.** We curate *native* Korean prompts from public Q&A and community websites via a two-step pipeline. **(1) Source discovery.** Using domain knowledge and targeted search, the authors compiled 54 candidate sites with user-posted questions and peer answers. Each site was assigned a license category: *A* (crawling and redistribution permitted), *B* (crawling allowed but commercial use and redistribution prohibited), and *C* (crawling prohibited). **(2) Legal triage and crawling.** We implement site-specific crawlers (one script per site) and exclude *C* sites, low-volume sources, heavily obfuscated structures, and near-duplicates. In this stage, we remove 26 websites from the list. Data from *B* sites is used for training and analysis but is not redistributed.

**Refinement and Filtering.** It is common practice to refine web-collected seed instructions either with templates or LLM rewriting (Mishra et al., 2021; Xu et al., 2023) prior to training. However, we observe that such normalization removes user artifacts (typos, abbreviations, mixed script, and

---

[4]We use the term language-mixing in the same sense as code-switching, that is, alternating between two or more languages within a single context.

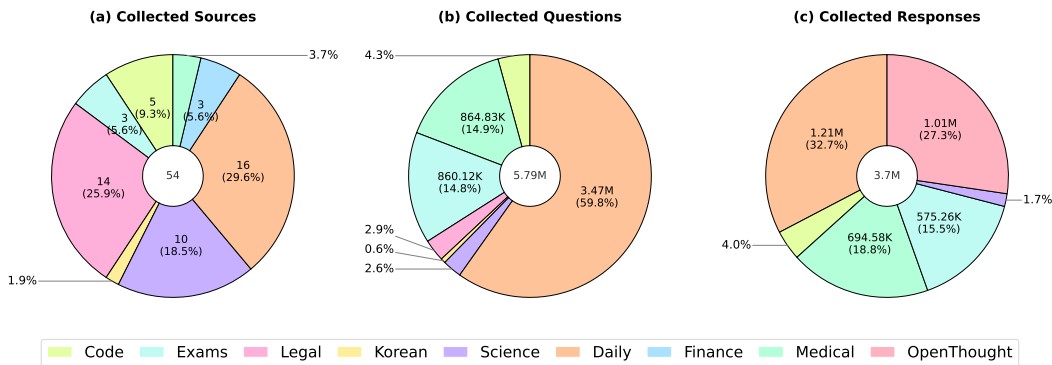

Figure 3: **Category distribution across different stages of the dataset collection.** (a) Sources (N=54): counts of the public Q&A and community websites we compiled; categories were manually assigned by the authors based on contextual review. (b) Questions: after crawling, items inherit the category from their source. (c) Responses: after response generation, we added OpenThought (Guha et al., 2025) as an additional source. Colors are shared across panels; centers show total counts.

internet style) that harm robustness at deployment, so we keep prompts verbatim. We apply only light automatic filters: discard prompts with a Korean-character ratio below 30%, and drop prompts that are too short or too long (length $< 50$ or $> 8{,}192$ characters). The Korean threshold was empirically chosen to exclude fully non-Korean items while retaining mixed-language coding prompts.

**Instruction Statistics.** Figure 3 illustrates details on the collected sources and prompts. Initially, roughly 25.9% of our compiled sources were legal websites. However, they tend to be small in scale or legally restricted, so they contribute only a minor share to the total number of crawled questions. In contrast, exam and daily communities host extensive, easily crawlable archives and are overrepresented. Given that long chain-of-thought training primarily improves reasoning capabilities rather than those tasks involving knowledge retrieval (Yeo et al., 2025; Sprague et al., 2024), we prioritize STEM/Code/Exam categories in subsequent collection and curation.

**Adding the OpenThought dataset.** Finally, our web-sourced training mix lacks competition-level problems that are known to cultivate reasoning ability (Guan et al., 2025). We therefore add prompts from OpenThought (Guha et al., 2025) by translations through *Gemini-2.5-Flash* (Comanici et al., 2025). Earlier attempts with *GPT-4o-mini* (Hurst et al., 2024), *Qwen2.5-72B-Instruct* (Qwen et al., 2025), and *Gemini-2.0-Flash* (Deepmind, 2024) produced training instabilities.

## 5.2 RESPONSE GENERATION

**SFT over Reinforcement Learning.** To build a strong Korean reasoning model, we focus on SFT in this work. Although recent studies report sizable gains from RL-based preference optimization (e.g., GRPO (Shao et al., 2024)), particularly for sub-32B models (Rastogi et al., 2025; Guo et al., 2025), these methods presume access to strong base models (Wang et al., 2025). For Korean, such strong seeds are scarce, making RL vulnerable to the cold-start problem with unstable reward learning and poor exploration (Shao et al., 2025). Consequently, we prioritize SFT with curated data to build a strong base model for subsequent RL efforts. Importantly, SFT has also been proven to be effective in training reasoning models (Hochlehnert et al., 2025; Ji et al., 2025), making it a reliable first step.

**Response Generation Methodology.** To build the SFT dataset, we initially consider two strategies: **(a) agreement-sampling**, where we sample multiple times from a teacher model and accept the first that an LLM judge (Zheng et al., 2023) deems consistent with the web-crawled answer, and **(b) hint-based refinement**, where we prepend the crawled answer and ask the model to refine it. However, we find both concerning: (a) is prohibitively compute-intensive; (b) risks leakage, artifacts, and distribution shift that can hurt generalization. Moreover, web-scraped answers are unreliable, and the recent strong LLMs have a chance to surpass crowd responses. It should also be noted that several works (Toshniwal et al., 2024), including OpenThought (Guha et al., 2025), and S1 (Muennighoff et al., 2025), have empirically shown that response filtering is not necessary, or hardly correlated with

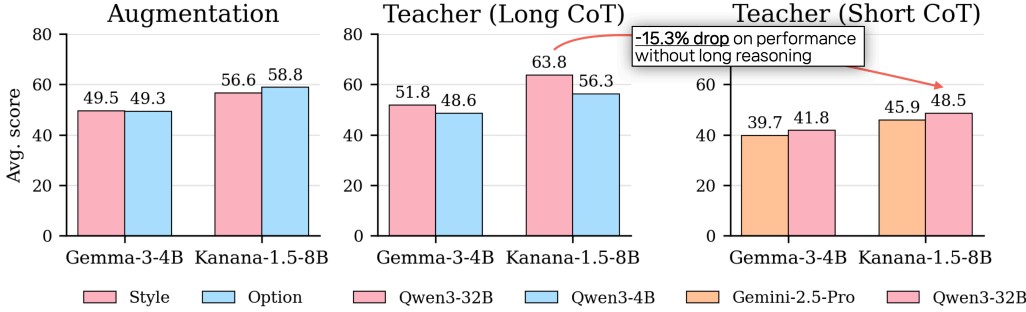

Figure 4: **Average scores across HAE-RAE Bench, MCLM, and KMMLU-Redux for Gemma-3-4B and Kanana-1.5-8B under three settings. (a) Augmentation.** Option and Style are comparable on Gemma-3-4B (49.3 vs 49.5), while Option has a modest edge on Kanana-1.5-8B (58.8 vs 56.6); neither augmentation is uniformly superior. **(b) Teacher (Long CoT).** Qwen3-32B yields higher averages than Qwen3-4B (Gemma: 51.8 > 48.6; Kanana: 63.8 > 56.3). **(c) Teacher (Short CoT).** With shot CoT, Qwen3-32B tops Gemini-2.5-Pro (Gemma: 41.8 > 39.7; Kanana: 48.5 > 45.9). **Overall, Language-Mixed CoT and using Qwen3-32B as the teacher provide the strongest gains; both augmentation choices offer benefits.**

the performance of the downstream model. We, therefore, choose to regenerate all targets from the prompt alone with a strong teacher, without any web-collected oracle.

**Selecting Response Format and Teacher Model.** To select the teacher model, we evaluate two candidates, *Qwen3-32B* and *Qwen3-4B*. We also test a *short-CoT* setting, where the model is trained on plain instructional responses without explicit reasoning traces, similar to conventional instruction-tuning outputs. This variant is implemented with *Qwen3-32B* (reasoning disabled) and *Gemini-2.5-Pro*. Figure 4 reports the downstream results across teachers. As expected, *Qwen3-32B* with language-mixed CoT delivers the strongest performance. Notably, *Qwen3-4B* with reasoning surpasses both *Gemini-2.5-Pro* and *Qwen3-32B* without reasoning, highlighting the importance of explicit reasoning in unlocking LLM capabilities. See Ablation A.4 for more details on training with different teachers.

**Format Augmentation.** The Exams category is highly standardized, typically a question with four options. To improve robustness beyond the fixed template, we apply two augmentations: **(a) Style augmentation.** We keep the question unchanged and prepend or append short stylistic directives.[5] **(b) Option augmentation**. We use a BM25 retriever (Robertson et al., 1995) over the exam pool to find similar questions and merge their distractor options with the original item. We drop items containing negation cues to avoid semantic flips, remove near-duplicate items, cap the merged list at 10 options, and preserve the original correct option as the gold label. As shown in Figure 4, training with either augmentation alone yields comparable performance, so we adopt both.

## 5.3  Dataset Composition

Building on these lessons, we use Qwen3-32B to generate language-mixed CoTs for the 5.79M prompts and augmentations. After filtering degenerations and enforcing Korean-ratio bounds, we obtain **Yi-Sang** with 3.7M long-reasoning trajectories. However, while scaling data generally improves performance, multi-epoch training on our full 3.7M instances is impractical due to limited compute budget. We therefore run targeted ablations to identify a smaller, high-yield mixture.

**What benefit does each category bring?** We begin by training on each category at a time. Each ablation run additionally includes 3,000 items from the Exams category to teach formatting. In Table 3, we observe that OpenThought delivers the largest gains on MCLM, followed by Science and Code. Exams are the most effective source for *HAE-RAE Bench* and *KMMLU-Redux*. Notably, the Medical category appears highly specialized: it boosts *ClinicalQA* but significantly hinders performance on all other benchmarks. These trends hold across both models, suggesting that the most effective

---

[5]Examples include: "return the final answer in \boxed{} format", or "output format: answer:<N>".

Table 3: **Contribution of individual training categories. OpenThought and Exams provide the largest gains, followed by Code and Science.** Medical boosts ClinicalQA but consistently harms performance on other benchmarks. Each run uses 50k examples from the target category, plus 3k EXAMS items for formatting. Since SCIENCE has only 37k examples, we use 37k+3k without up-sampling. The highest-scoring model is highlighted in green and the lowest-scoring model in red. Abbreviations: Clin. = ClinicalQA.

| Category | Gemma-3-4B | | | | Kanana-1.5-8B | | | |
|---|---|---|---|---|---|---|---|---|
| | HRB | MCLM | KMMLU-R | Clin. | HRB | MCLM | KMMLU-R | Clin. |
| OpenThought | 54.9 | **55.8** | 53.0 | 62.1 | **74.6** | **57.4** | 64.4 | **73.97** |
| Daily | 54.2 | 34.9 | 51.9 | 62.4 | 69.1 | 36.4 | 58.7 | 70.0 |
| Medical | 50.5 | 20.9 | 49.4 | **65.6** | 64.5 | 28.7 | 57.0 | 70.3 |
| Code | 53.5 | 38.8 | 51.5 | 59.0 | 69.4 | 38.0 | 59.1 | 64.4 |
| Exams | **56.4** | 27.9 | **64.2** | 60.0 | 69.5 | 33.3 | **67.0** | 69.9 |
| Science | 52.2 | 37.2 | 52.1 | 61.9 | 68.3 | 41.1 | 58.8 | 67.5 |

mixture uses OpenThought and Exams as the foundation and adds Science/Code for additional math robustness. Additionally, we find that scaling the Daily and Medical subsets has adverse effects on overall performance, leading us to exclude them from the final training composition. [6]

**Finalizing the dataset.** Finally, we decide to train only with: OpenThought, Code, Exams, and Science. This approximates about 1.8M instances. To surface data issues, we conduct a one-epoch shakedown run with a proxy model (Kanana-1.5-2.1B). We define a "loss spike" as an abrupt rise in loss that does not recover immediately in the subsequent step. When such spikes occur, we locate the batch, manually inspect the items, implement a rule-based filter to remove the failure pattern from the entire dataset, and restart the run. This process is repeated until the loss curve stabilizes. During this process, we identify three recurring triggers: degeneration cases where responses endlessly repeat identical phrases; samples that contain multiple `<think> ... </think>` blocks; and instances in which the final solution after the `</think>` tag is written in a non-Korean language. We also find that a small number of extremely long reasoning traces disproportionately slow training, leading us to discard any instance exceeding 16k tokens.

**Decontamination** We decontaminate the training corpus against both held-in and held-out benchmarks using a 13-gram overlap filter applied to prompts and reasoning traces. Before constructing $n$-grams, we perform morphological segmentation and normalization with MeCab-KO (MeCab-KO Contributors). We then run two passes: (i) build 13-grams over the normalized strings and (ii) build 13-grams over the raw text; any training instance that shares at least one 13-gram with any benchmark item in either pass is removed, effectively eliminating exact and near duplicates. We intentionally avoid embedding-based decontamination, as exhaustive semantic matching over 3.7M trajectories and nine benchmarks would be computationally costly and risks discarding legitimate background knowledge rather than true leakage. Overall, the 13-gram decontamination removes about $0.7\%$ of trajectories ($\sim$25.9k). After all filtering steps, the finalized **YI-SANG-HQ** corpus contains **260k** instances, composed of **62k** from OpenThought, **86k** from Code, **37k** from Science, and **66k** from Exams.

## 6 RESULTS

**YI-SANG-HQ Achieves State-Of-The-Art Performance.** **KO-REAson-35B**, based on A.X-3.1 and trained on YI-SANG-HQ, outperforms state-of-the-art reasoning models of comparable scale, including *GPT-OSS-20B* (Agarwal et al., 2025b), *DeepSeek-R1-32B* (Guo et al., 2025), *EXAONE-Deep-32B* (Research et al., 2025), and *QwQ-32B* (Team, 2025). In Table 4, across nine benchmarks, **KO-REAson-35B** achieves the best performance on five tasks and ranks second on the remaining four, achieving the highest overall average. We note that performance on competition-level math datasets (AIME2024, KSM) trails *GPT-OSS-20B*, which we attribute to the relatively small amount of competition-style reasoning data in our mixture: only $\sim$ 60k translated OpenThought items after filtering, compared to nearly 1M competition-style problems in the original OpenThought project and $\sim$ 0.5M in Liu et al. (2025b). Nevertheless, **KO-REAson-35B** ranks second place on both benchmarks while relying primarily on web-collected data that constitutes the majority of training.

---

[6] See Section A.4 for details.

Table 4: **Comparison of 20B+ reasoning models. KO-REAson-35B (ours) matches state-of-the-art peers of similar scale while using only openly available data and code.** Entries are reported as mean$_{SE}$ over $n = 3$ independent runs. **Bold** marks the row-best; underline marks the second-best. When standard-error intervals overlap, ties are co-highlighted. Exact prompts are provided in the supplementary materials. Math (Ko) and AIME24 (Ko) are subsets of MCLM.

| Category | Benchmark | GPT-OSS-20B | DS-R1-32B | EXAONE-Deep-32B | QwQ-32B | KO-REAson-35B |
|---|---|---|---|---|---|---|
| General | KMMLU-Redux | $67.6_{0.1}$ | $70.0_{1.6}$ | $68.2_{2.2}$ | $\underline{74.7}_{1.0}$ | $\mathbf{76.0}_{0.4}$ |
| | KMMLU-Pro | $42.9_{0.5}$ | $45.7_{0.3}$ | $43.5_{1.8}$ | $\mathbf{51.0}_{1.1}$ | $\underline{47.4}_{0.6}$ |
| | KMMLU-Hard | $39.0_{0.2}$ | $43.3_{1.0}$ | $43.5_{1.9}$ | $\underline{49.0}_{1.0}$ | $\mathbf{51.4}_{0.5}$ |
| Reasoning | Math (Ko) | $82.8_{1.7}$ | $\underline{85.4}_{2.1}$ | $84.8_{2.9}$ | $82.3_{0.7}$ | $\mathbf{87.5}_{0.6}$ |
| | AIME2024 (Ko) | $\mathbf{71.1}_{6.9}$ | $\underline{51.7}_{7.1}$ | $58.3_{11.8}$ | $53.3_{9.4}$ | $\underline{66.7}_{11.5}$ |
| | KSM | $\mathbf{72.1}_{4.7}$ | $62.8_{5.1}$ | $\underline{65.7}_{17.9}$ | $60.5_{14.4}$ | $\underline{65.7}_{8.6}$ |
| Ko-Specific | HRB | $65.1_{0.7}$ | $70.8_{0.4}$ | $\underline{76.1}_{0.3}$ | $75.5_{1.1}$ | $\mathbf{78.9}_{0.7}$ |
| | CLIcK | $57.2_{0.7}$ | $66.6_{0.6}$ | $\underline{67.6}_{0.3}$ | $\mathbf{70.9}_{0.6}$ | $\mathbf{70.9}_{0.3}$ |
| | KoBALT-700 | $31.0_{1.4}$ | $33.3_{0.1}$ | $32.6_{5.6}$ | $\mathbf{37.7}_{2.0}$ | $\underline{34.9}_{1.0}$ |
| **Average** | | $58.8_{1.2}$ | $56.4_{4.5}$ | $57.4_{5.2}$ | $\underline{59.6}_{3.1}$ | $\mathbf{64.0}_{2.5}$ |

Table 5: **Performance of nine models (4B–35B) trained on YI-SANG-HQ. The benefits of YI-SANG-HQ are consistent across model families and parameter scales.** Results are mean$_{SE}$ over $n = 3$ independent runs. Cases where performance drops after training (without overlap of standard errors) are highlighted. Abbreviations: K.M.-R = KMMLU-Redux; K.M.-P = KMMLU-Pro; K.M.-H = KMMLU-Hard.

| Model | K.M.-R | K.M.-P | K.M.-H | MATH | AIME24 | KSM | HRB | CLIcK | KoBALT |
|---|---|---|---|---|---|---|---|---|---|
| | | | | *<5B Models* | | | | | |
| Gemma-3-4B | $40.7_{1.7}$ | $26.7_{1.3}$ | $19.4_{0.2}$ | $41.9_{25.0}$ | $1.7_{2.4}$ | $12.8_{6.7}$ | $49.1_{5.2}$ | $45.9_{4.0}$ | $12.0_{2.9}$ |
| + YI-SANG-HQ | $65.5_{3.5}$ | $35.3_{3.6}$ | $41.6_{3.2}$ | $69.7_{1.4}$ | $15.0_{2.4}$ | $38.8_{13.9}$ | $61.0_{9.9}$ | $55.2_{5.1}$ | $20.0_{4.1}$ |
| | | | | *<10B Models* | | | | | |
| Qwen-2.5-7B | $52.6_{0.3}$ | $34.0_{0.1}$ | $20.7_{0.2}$ | $58.1_{6.4}$ | $6.7_{0.0}$ | $15.8_{0.3}$ | $60.4_{1.1}$ | $56.9_{0.6}$ | $19.3_{0.8}$ |
| + YI-SANG-HQ | $72.0_{0.4}$ | $44.6_{0.5}$ | $46.7_{0.1}$ | $77.3_{0.7}$ | $41.7_{11.8}$ | $49.7_{1.5}$ | $65.0_{0.8}$ | $61.0_{0.4}$ | $24.1_{1.4}$ |
| A.X-3.1-7B | $62.4_{0.5}$ | $38.8_{0.3}$ | $36.3_{2.0}$ | $48.2_{18.9}$ | $34.6_{39.6}$ | $17.3_{3.5}$ | $71.3_{0.7}$ | $64.8_{0.4}$ | $25.0_{2.3}$ |
| + YI-SANG-HQ | $70.0_{0.9}$ | $39.0_{0.7}$ | $45.7_{0.5}$ | $82.8_{2.9}$ | $33.3_{14.1}$ | $53.4_{1.3}$ | $72.5_{0.9}$ | $62.0_{0.9}$ | $23.9_{0.7}$ |
| KONI-Llama-3.1-8B | $20.7_{0.4}$ | $16.0_{0.6}$ | $9.7_{0.4}$ | $18.7_{2.1}$ | $3.3_{0.0}$ | $4.8_{0.2}$ | $21.7_{1.8}$ | $21.9_{0.4}$ | $0.5_{0.2}$ |
| + YI-SANG-HQ | $69.6_{0.1}$ | $39.6_{0.5}$ | $44.7_{0.6}$ | $71.7_{1.4}$ | $31.7_{7.1}$ | $38.3_{0.4}$ | $58.3_{0.8}$ | $56.5_{1.0}$ | $21.4_{0.4}$ |
| Llama-3.1-8B | $40.1_{1.2}$ | $23.8_{1.3}$ | $19.5_{0.2}$ | $29.3_{7.1}$ | $1.7_{2.4}$ | $5.1_{0.2}$ | $43.7_{0.4}$ | $41.5_{0.6}$ | $8.1_{0.6}$ |
| + YI-SANG-HQ | $68.9_{0.2}$ | $38.6_{0.4}$ | $45.3_{0.0}$ | $72.2_{0.7}$ | $26.7_{14.1}$ | $38.7_{0.7}$ | $54.9_{0.4}$ | $54.9_{0.4}$ | $18.9_{0.1}$ |
| Kanana-1.5-8B | $53.7_{4.9}$ | $37.7_{0.2}$ | $27.2_{0.1}$ | $54.5_{0.0}$ | $10.0_{0.0}$ | $15.0_{0.1}$ | $70.3_{8.2}$ | $63.5_{0.3}$ | $20.6_{0.5}$ |
| + YI-SANG-HQ | $70.7_{0.5}$ | $39.9_{0.7}$ | $44.8_{0.5}$ | $67.7_{1.4}$ | $30.0_{0.0}$ | $39.8_{2.1}$ | $72.9_{0.9}$ | $64.0_{0.4}$ | $28.6_{0.5}$ |
| | | | | *<20B Models* | | | | | |
| Gemma-3-12B | $59.1_{0.6}$ | $39.9_{0.3}$ | $29.8_{0.0}$ | $73.2_{2.1}$ | $15.0_{7.1}$ | $28.1_{0.3}$ | $69.8_{0.2}$ | $62.2_{0.5}$ | $26.0_{0.1}$ |
| + YI-SANG-HQ | $72.7_{0.9}$ | $43.2_{1.2}$ | $47.1_{0.1}$ | $75.3_{0.7}$ | $35.0_{7.1}$ | $46.1_{6.7}$ | $68.8_{0.2}$ | $64.6_{0.2}$ | $29.6_{0.1}$ |
| Qwen-2.5-14B | $24.4_{1.3}$ | $22.7_{0.6}$ | $14.0_{1.0}$ | $64.6_{2.9}$ | $8.3_{2.4}$ | $20.7_{0.8}$ | $20.7_{9.3}$ | $31.5_{0.7}$ | $19.8_{0.8}$ |
| + YI-SANG-HQ | $77.1_{0.7}$ | $50.0_{0.2}$ | $51.5_{0.6}$ | $81.8_{1.4}$ | $38.3_{2.4}$ | $55.6_{8.4}$ | $74.5_{0.2}$ | $67.5_{0.5}$ | $34.5_{1.3}$ |
| | | | | *<30B Models* | | | | | |
| A.X-3.1-35B | $72.2_{0.2}$ | $47.3_{0.7}$ | $44.2_{0.2}$ | $73.1_{2.1}$ | $16.7_{3.3}$ | $26.8_{0.8}$ | $84.0_{0.5}$ | $76.6_{0.6}$ | $35.5_{0.2}$ |
| + YI-SANG-HQ | $76.0_{0.4}$ | $47.4_{0.6}$ | $51.4_{0.5}$ | $84.5_{0.6}$ | $66.7_{11.5}$ | $65.7_{8.6}$ | $78.9_{0.7}$ | $70.9_{0.3}$ | $34.9_{1.0}$ |

This underscores the quality of the newly collected user prompts. We leave to future work on incorporating larger volumes of translated competition-style data to push performance further.

**YI-SANG-HQ Demonstrates Persistent Gains Across Model Size and Family.** To further validate the efficacy of **YI-SANG-HQ** across diverse settings, we train nine models spanning 4B to 35B parameters from six different model families. Improvements are consistent across both scale and architecture, with especially pronounced gains on math-intensive benchmarks such as *Math (Ko)*, *AIME2024*, and *KSM*, where models of all sizes benefit substantially. Korean-specific tasks (*HRB1.0*, *CLIcK*, and *KoBALT-700*) also show steady improvements, underscoring the value of YI-SANG-HQ's curated multilingual and culturally grounded data. General knowledge evaluations (*KMMLU-Redux*, *KMMLU-Pro*, *KMMLU-Hard*) likewise improve, further demonstrating the broad coverage of the dataset. Performance degradation is observed in only two cases, both with marginal drops of less than two points. Overall, YI-SANG-HQ proves to be a versatile and widely applicable training resource,

capable of boosting models across families and scales, and offering substantial value for future research in multilingual and reasoning-focused LLMs.

**Cross-Lingual and Multi-Modal Free Lunch.** To investigate whether post-training on Yᴵ-Sᴀɴɢ-HQ yields broader generalization, we evaluate Gemma3-12B and its post-trained variant (KO-REAson-12B) on two English reasoning benchmarks (AIME-2025, short-form math; GPQA, STEM MCQA (Rein et al., 2024)) and two Korean vision-language benchmarks (KAIO-2, short-form STEM reasoning (Lee et al., 2025b); HAERAE-Vision, long-form commonsense reasoning). KO-REAson-12B outperforms the base model on all four, indicating both cross-lingual and multimodal gains. We attribute the English improvements to two factors: (i) the benchmarks emphasize largely universal math and science knowledge, which facilitates transfer across languages, and (ii) our **Language-Mixed CoT** includes English reasoning steps that push general reasoning capabilities. English performance gains are consistent over all trained models, see Table 14 for more details. Additionally, we also observe gains on visual reasoning despite no image data in post-training, consistent with prior reports of a "multi-modal free lunch." (Choi et al., 2024; Rastogi et al., 2025) However, the transfer appears selective, with strong gains on reasoning-heavy tasks and limited benefit on shallow, factoid-style evaluations. See Appendix D.2 for complete results and experimental details.

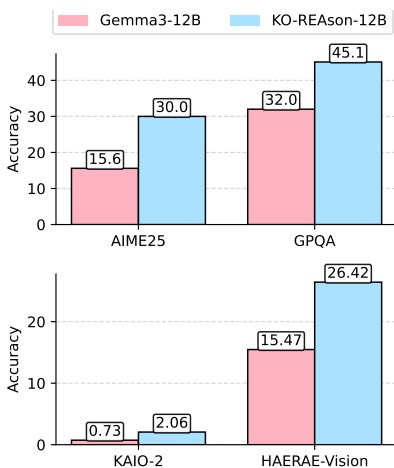

Figure 5: **Performance of Gemma3-12B and its post-trained variant on English reasoning benchmarks and Korean multimodal benchmarks.** KO-REASON-12B, trained only with text supervision, shows consistent gains across all tasks, indicating both cross-lingual and multimodal transfer.

## 7 Conclusion

In this work, we present practical recipes for building reasoning models for mid-resource languages through a Korean case study. We introduce Language-Mixed CoT and curate 5.9M native-authored Korean prompts, underscoring the value of better supervision signals and high-quality local data. Using Qwen3-32B as the teacher, we construct and release Yᴵ-Sᴀɴɢ, the largest publicly available Korean training resource. Its high-yield subset, Yᴵ-Sᴀɴɢ-HQ, delivers consistent gains in general knowledge and reasoning across six model families spanning 4B–35B parameters, rivaling models trained on proprietary data. We hope our work benefits Korean practitioners and the broader multilingual community, offering guidance for training their own reasoning LLMs.

## Acknowledgments

This research was supported by the Korea Institute of Science and Technology Information (KISTI) in 2026 (No. (KISTI)K26L3M1C1), aimed at developing KONI (KISTI Open Neural Intelligence), a large language model specialized in science and technology. This work was partly supported by an Institute of Information & communications Technology Planning & Evaluation (IITP) grant funded by the Korean Government (MSIT) (No. RS-2021-II211343, Artificial Intelligence Graduate School Program (Seoul National University)), the National Research Foundation of Korea(NRF) grant funded by the MSIT (RS-2024-00354218)

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

# A    ADDITIONAL DETAILS ON YI-SANG.

## A.1    ORIGIN

Our dataset takes its name from Yi Sang (1910-1937; pen name of Kim Hae-gyeong), a Korean modernist and architect, known for his mathematically inflected literature. He employed geometric notation, numerical sequences, and experimental layouts into Korean literacy works. The name reflects our focus on formal Korean reasoning. Yi Sang also echoes a Korean noun, meaning "the most complete state," consistent with our goal to create the strongest reasoning dataset.

## A.2    PROMPTS

Figure 6 presents the system prompt used throughout the paper to generate **Language-Mixed CoT** from teacher models. We notice that longer and more detailed instructions are likely to constrain stylistic diversity of responses. Therefore, we keep the prompt as simple as possible.

> Think carefully, do not translate the question while solving. Preserve the question in Korean so that you keep all details without adding noise. After you finish thinking, state your answer in fluent and coherent Korean.

Figure 6: System prompt used for dataset generation.

## A.3    LICENSE

In Table 6 we detail the license of our trained models. The models will be made available on HuggingFace. Both datasets YI-SANG and YI-SANG-HQ will be made available under the MIT License.

Table 6: **Summary of Base models, upstream licenses, our trained model names, and release licenses.** We resort to the most open license possible.

| Base Model | Upstream License | Trained Model (ours) | Release License |
|---|---|---|---|
| Gemma3-4B | Gemma License | KO-REAson-G3-4B-0831 | Gemma License |
| Gemma3-12B | Gemma License | KO-REAson-G3-12B-1002 | Gemma License |
| Llama-3.1-8B | Llama3 Community License | KO-REAson-L3_1-8B-0831 | Llama3 Community License |
| KONI-Llama-3.1-8B | Llama3 Community License | KO-REAson-KL3_1-8B-0831 | Llama3 Community License |
| A.X-3.1-Light | Apache 2.0 | KO-REAson-AX3_1-8B-0831 | Apache 2.0 |
| A.X-3.1 | Apache 2.0 | KO-REAson-AX3_1-35B-1002 | Apache 2.0 |
| Qwen2.5-7B | Apache 2.0 | KO-REAson-Q2_5-7B-0831 | Apache 2.0 |
| Qwen2.5-14B | Apache 2.0 | KO-REAson-Q2_5-14B-1002 | Apache 2.0 |
| Kanana1.5-8B | Apache 2.0 | KO-REAson-K2505-8B-0831 | Apache 2.0 |

## A.4    ADDITIONAL ABLATIONS

**Training with different Teachers.**    To investigate whether our LM-CoT distillation pipeline is agnostic to the choice of teacher, we apply the same procedure using both DEEPSEEK-R1-32B and QWEN3-32B as teachers, and distill into two students: KANANA-1.5-8B and GEMMA3-4B. Table 7 summarizes the results. For KANANA-1.5-8B, supervision from DEEPSEEK-R1-32B improves HAE-RAE Bench / MCLM / KMMLU-R from 60.8 / 45.7 / 48.1 to 71.0 / 48.8 / 58.9, while QWEN3-32B yields even larger gains (74.6 / 57.4 / 64.4). For GEMMA3-4B, DEEPSEEK-R1-32B provides modest improvements, mainly on MCLM and KMMLU-R, and QWEN3-32B again delivers the strongest student, raising performance to 54.9 / 55.8 / 53.0. These trends indicate that our pipeline consistently benefits different base models and teachers, while the absolute student performance remains bounded by the capability of the chosen teacher.

Table 7: **Teacher and student performance on HAE-RAE Bench, MCLM, and KMMLU-R.** Both teachers DeepSeek-R1-32B (DS) and Qwen3-32B (Q3) yield performance gains, proportionate to their original performance.

| Models | HAE-RAE Bench | MCLM | KMMLU-R |
|---|---|---|---|
| **Student Model Performance** | | | |
| (Base) Kanana-1.5-8B | 60.8 | 45.7 | 48.1 |
|    Supervised by DS-R1-32B | 71.0 | 48.8 | 58.9 |
|    Supervised by Q3-32B | 74.6 | 57.4 | 64.4 |
| (Base) Gemma3-4B | 53.5 | 43.4 | 38.7 |
|    Supervised by DS-R1-32B | 53.3 | 45.7 | 49.6 |
|    Supervised by Q3-32B | 54.9 | 55.8 | 53.0 |
| **Teacher Model Performance** | | | |
| DeepSeek-R1-32B | 71.8 | 75.2 | 70.2 |
| Qwen3-32B | 75.7 | 83.7 | 81.0 |

**Scaling the Medical subset.** To test for emergent gains, we double the Medical subset from 50k to 100k and retrain. Table 8 reports the performance gains relative to 50k. Gemma-3-4B decreases on all benchmarks, with the largest drop on ClinicalQA. Kanana-1.5-8B exhibits near-zero changes. Therefore, we exclude the Medical category from the final training mixture.

Table 8: **Size ablation on the Medical subset. Doubling the Medical subset from 50k to 100k leads to negative performance effects.** Reported values show the change in accuracy; Avg (non-Clin.) is the unweighted mean of non-clinical benchmarks.

| Model | $\Delta$ Avg (non-Clin.) | $\Delta$ ClinicalQA |
|---|---|---|
| Gemma-3-4B | $-0.76$ | $-2.30$ |
| Kanana-1.5-8B | $+0.09$ | $+0.10$ |

**Scaling the Daily subset.** Table 3 shows that Daily rarely leads any benchmark. We scale Daily by $s \in \{20, 50, 100\}$k and mix 15k instances each from OpenThought and Exams. The two datasets are added to prevent downstream models from showing deflated scores on the academic benchmarks, since the Daily category is likely to lack academic value. As reported in Table 9, performance consistently drops as we scale. Therefore, we also exclude the Daily category.

Table 9: **Size ablation on the Daily subset. Overall performance declines as the subset increases in size.** This may partly stem from limited benchmark coverage; nonetheless, the evidence is not enough to tolerate consistent drops across the remaining benchmarks. The highest-scoring model is highlighted in **green**.

| Data Mix | Gemma-3-4B | | | Kanana-1.5-8B | | |
|---|---|---|---|---|---|---|
| | HRB | MCLM | KMMLU-R | HRB | MCLM | KMMLU-R |
| 2:1.5:1.5 | **56.2** | **48.8** | **54.0** | **73.1** | **48.8** | **60.8** |
| 5:1.5:1.5 | 55.5 | 48.1 | 53.0 | 68.9 | 45.7 | 59.7 |
| 10:1.5:1.5 | 55.8 | 40.3 | 51.6 | 69.9 | 43.4 | 58.8 |

**Scaling to a bigger dataset.** We also investigate the effect of scaling to a larger training set. We conduct a controlled experiment with a subset of 780k samples, consisting of YiSang-HQ combined with 500k English OpenThought instances (denoted as YiSang-HQ+OT(en)), and fine tuned Gemma3-4B on this mixture.

Table 10: Effect of adding 500k English OpenThought samples on benchmark performance.

| Benchmark | YiSang-HQ | YiSang-HQ+OT(en) |
|---|---|---|
| KMMLU-Redux | 65.3 | 63.5 |
| HAE-RAE Bench | 61.0 | 59.4 |
| MCLM-Ko | 55.0 | 67.4 |

As shown in Table 10, adding a large amount of English OpenThought data leads to a substantial improvement on MCLM-Ko (from 55.0 to 67.4), a math focused benchmark that particularly benefits from additional Olympiad style problems. However, this comes at the cost of lower performance on KMMLU-Redux and HAE-RAE, both of which contain a significant amount of Korean specific content. In other words, the larger and more math heavy mixture shifts the model toward stronger mathematical reasoning, while slightly degrading its overall Korean performance. Given that our primary objective is to build a balanced model for Korean usage, rather than optimizing a specific subset of reasoning benchmarks, we chose not to adopt this larger mixture in the final training runs. Nevertheless, for applications that prioritize reasoning performance in math and related benchmarks, extending the training data with additional English OpenThought style problems, as in YiSang-HQ+OT(en), appears to be a promising direction.

## A.5 ABLATION DETAILS

Table 11 and 12 provide detailed results behind Figure 4.

Table 11: **Comparison of two augmentation strategies (style and option); no single method demonstrates a clear advantage.** The highest-scoring model is highlighted in green.

| Augmentation | Gemma-3-4B | | | Kanana-1.5-8B | | |
|---|---|---|---|---|---|---|
| | HRB | MCLM | KMMLU-R | HRB | MCLM | KMMLU-R |
| Style | **56.4** | 27.9 | **64.2** | 69.5 | 33.3 | **67.0** |
| Option | 55.8 | **30.2** | 61.9 | **72.8** | **37.2** | 66.5 |

Table 12: **Comparison of different teacher models and response formats. Training on long chain-of-thought reasoning generated by Qwen3-32B shows the best performance.** Performance caps are most pronounced in the MCLM benchmark, implying its effectiveness in boosting reasoning performance. The highest-scoring model is highlighted in green.

| Teacher Model | Gemma-3-4B | | | Kanana-1.5-8B | | |
|---|---|---|---|---|---|---|
| | HRB | MCLM | KMMLU-R | HRB | MCLM | KMMLU-R |
| *Language-Mixed CoT* | | | | | | |
| Qwen3-32B | **54.4** | **48.1** | **53.0** | **73.1** | **57.4** | **60.8** |
| Qwen3-4B | 48.6 | 45.0 | 52.3 | 67.8 | 41.9 | 59.1 |
| *Solution Only (Short CoT)* | | | | | | |
| Gemini-2.5-Pro | 49.5 | 25.6 | 44.1 | 67.6 | 24.0 | 46.2 |
| Qwen3-32B | 51.3 | 28.7 | 45.3 | 68.5 | 23.3 | 53.7 |

# B ADDITIONAL DETAILS ON MODEL TRAINING.

## B.1 MODELS

**Gemma-3** (Team et al., 2025) is Google's third-generation open model family. We use 4B and 12B instruction-tuned variants. Gemma-3 is a multimodal model (text and vision), though in this work we use it purely for text. The 4B version is pretrained on roughly 4T tokens, and the 12B on about 12T tokens. It is massively multilingual, covering more than 140 languages without a special focus on any single one.

**Qwen-2.5** (Qwen et al., 2025) is built by Alibaba Cloud and trained on up to 18T tokens. It is grounded primarily in Chinese and English, but demonstrates solid multilingual capabilities with decent coverage of Korean (Hong et al., 2025). In our experiments, we use both the 7B and 14B instruction-tuned variants.

**A.X-3.1**    (Lab, 2025) is a family of LLaMA-style models developed by SK Telecom with a particular focus on Korean. It is trained on approximately 2.1 trillion tokens and achieves top-tier scores on Korean benchmarks, such as KMMLU, while still performing well in English. We employ both the 8B and 35B variants.

**Kanana-1.5-8B**    (Bak et al., 2025) is a bilingual English–Korean model, trained by Kakao, with an 8B parameter LLaMA-style transformer. It is trained on about 3T tokens, with more than 10% Korean content, while the rest is primarily English. The training recipe includes staged pretraining and efficiency optimizations.

**Llama-3.1-8B-Instruct**    (Grattafiori et al., 2024) is trained on approximately 15T tokens and designed as a multilingual model but with emphasis on eight major languages, including English, German, French, Italian, Portuguese, Hindi, Spanish, and Thai. Although it is broadly multilingual, it remains relatively English-centric.

**KONI-Llama-3.1-8B**    (KISTI, 2024) is a continual pretrained variant of Llama-3.1 developed by KISTI. It starts from the base Llama-3.1-8B architecture and undergoes continued pretraining on 0.5 trillion tokens of additional Korean text and domain-specific corpora in science and technology.

### B.2 Hyperparameters

Training hardware spans from eight NVIDIA H100 to twenty-four NVIDIA H200 GPUs. Ablations use 5 epochs, a global batch size of 128, bfloat16 precision, and AdamW (learning rate $2 \times 10^{-5}$ with 10% warmup; weight decay $1 \times 10^{-5}$). Loss is computed only on reasoning traces and solutions. We employ PyTorch FSDP, Liger kernels (Hsu et al., 2024), and FlashAttention-2 (Dao et al., 2022). For the final runs on YI-SANG-HQ we scale the global batch size to 512.

### B.3 Packing

)We train Gemma-3-4B and Kanana-1.5-8B on YI-SANG-HQ under two settings (with vs. without packing). Although packing provided substantial speedups, as shown in Table 13 we observe measurable drops on general-knowledge and reasoning benchmarks; accordingly, all reported models are trained without packing.

| Benchmarks | Gemma-3-4B | | Kanana-1.5-8B | |
|---|---|---|---|---|
| | w packing | wo packing | w packing | wo packing |
| KMMLU-Redux | 62.87 | **64.19** | 70.06 | **71.30** |
| HAE-RAE Bench | **59.62** | 55.33 | **75.88** | 73.73 |
| MCLM-Ko | 55.04 | **58.91** | 62.02 | **65.12** |
| Training Time | 576 | 1728 | 1296 | 3360 |

Table 13: Comparison of model performance with and without packing.

## C    Additional Details on Evaluation

### C.1 Prompts

Figure 7 is the prompt used for evaluation.

> 문제 풀이를 마친 후, 최종 정답을 다음 형식으로 작성해 주세요: \boxed{N}.

Figure 7: System prompt used for evaluation on Korean benchmarks.

## C.2 Processing Details

We extract each model's final answer from the first `\boxed{...}`, that appears after the model's hidden "think" (reasoning) content. Any `\boxed{...}`, strings that occur inside the think section are ignored. If multiple `\boxed{...}`, entries appear in the visible answer, we always take the first one and disregard the rest, even if they contradict one another. An answer is credited only if this first post-think `\boxed{...}`, is parsable. If the model fails to produce a parsable `\boxed{...}`,, the response is marked *incorrect*, even when the correct value appears elsewhere in plain text.

If a generation runs to the maximum token limit and no parsable `\boxed{...}`, is produced (typically due to degeneration), the item is marked *incorrect*. By contrast, if a generation is interrupted before reaching the max token limit due to a hardware or runtime failure, we re-run the same prompt once with the same decoding settings; the score is based on the retry.

# D Additional Results

## D.1 Cross-Lingual Gains on English Benchmarks

Table 14: **Performance of nine models (4B–35B) trained on Yi-Sang-HQ.** Results are mean$_{\text{SE}}$ over $n=3$ runs on AIME24, AIME25, and GPQA. The benefits of Yi-Sang-HQ are consistent across model families and scales.

| Model | AIME24 | AIME25 | GPQA |
|---|---|---|---|
| *<5B Models* | | | |
| Gemma-3-4B | $6.7_{5.8}$ | $10.0_{8.8}$ | $19.5_{2.9}$ |
| + Yi-Sang-HQ | $23.3_{17.3}$ | $22.2_{6.9}$ | $32.2_{7.7}$ |
| *<10B Models* | | | |
| Qwen-2.5-7B | $6.7_{0.0}$ | $7.8_{1.9}$ | $27.1_{3.5}$ |
| + Yi-Sang-HQ | $41.1_{1.9}$ | $34.4_{3.8}$ | $43.1_{1.3}$ |
| A.X-3.1-7B | $13.3_{0.0}$ | $13.3_{5.8}$ | $25.6_{3.7}$ |
| + Yi-Sang-HQ | $46.7_{5.8}$ | $31.1_{1.9}$ | $37.7_{2.9}$ |
| KONI-Llama-3.1-8B | $0.0_{0.0}$ | $0.0_{0.0}$ | $14.1_{1.5}$ |
| + Yi-Sang-HQ | $21.1_{1.9}$ | $32.2_{1.9}$ | $39.2_{0.8}$ |
| Llama-3.1-8B | $0.0_{0.0}$ | $0.0_{0.0}$ | $19.2_{0.5}$ |
| + Yi-Sang-HQ | $28.9_{3.8}$ | $21.1_{1.9}$ | $40.2_{0.8}$ |
| Kanana-1.5-8B | $5.6_{1.9}$ | $12.2_{1.9}$ | $31.1_{2.5}$ |
| + Yi-Sang-HQ | $25.6_{7.7}$ | $27.8_{1.9}$ | $38.9_{0.5}$ |
| *<20B Models* | | | |
| Gemma-3-12B | $13.3_{0.0}$ | $15.6_{1.9}$ | $32.0_{1.2}$ |
| + Yi-Sang-HQ | $42.2_{7.7}$ | $30.0_{5.8}$ | $45.1_{6.7}$ |
| Qwen-2.5-14B | $7.8_{5.1}$ | $13.3_{3.3}$ | $26.3_{1.3}$ |
| + Yi-Sang-HQ | $41.1_{15.0}$ | $42.2_{10.2}$ | $51.7_{6.6}$ |
| *<30B / 35B Models* | | | |
| A.X-3.1-35B | $15.6_{3.8}$ | $15.6_{1.9}$ | $37.0_{0.8}$ |
| + Yi-Sang-HQ | $58.9_{16.4}$ | $53.3_{12.0}$ | $47.8_{5.3}$ |

Alongside the results in Table 5, we also observe consistent gains on English reasoning benchmarks such as AIME2024/2025 and GPQA (Rein et al., 2024). While the improvements are not yet sufficient to rival state-of-the-art systems of similar scale, it is notable that every model improves across all English benchmarks despite never seeing English prompts during training. We attribute this to two factors. First, the math and science benchmarks used here largely test universal knowledge, making them less dependent on the training language and enabling transfer from the Korean supervision. Second, the proposed Language-Mixed CoT likely helps models maintain alignment with their original English distribution, since they continue to practice reasoning partly in English. These findings highlight promising directions for further study on cross-lingual transfer in reasoning.

## D.2 Cross-Modal Gains on Visual Language Benchmarks

Earlier works have discovered the "multi-modal free lunch", noting that Visual Language Models (VLMs) trained with text-only reasoning data often improve across a wide range of vision benchmarks (Choi et al., 2024; Li et al., 2025b). We extend this line of inquiry by evaluating Gemma3-12B, a model with a visual encoder but trained solely on YI-SANG-HQ, across three multimodal benchmarks: K-Viscuit (knowledge-focused, MCQA) (Park et al., 2024), HAERAE-Vision (reasoning, long-form) (Choi et al., 2026), and KAIO-2 (STEM/reasoning, short-form) (Lee et al., 2025b). As shown in Table 8, KO-REAson-12B achieves notable gains on reasoning-oriented tasks despite lacking vision training. Unlike prior reports of across-the-board improvements (Rastogi et al., 2025), however, we find that shallow factoid-style benchmarks such as K-Viscuit see little to no benefit. This suggests that the free lunch of text-based reasoning transfers selectively: boosting reasoning-heavy multi-modal tasks, but not those requiring surface-level factual recall.

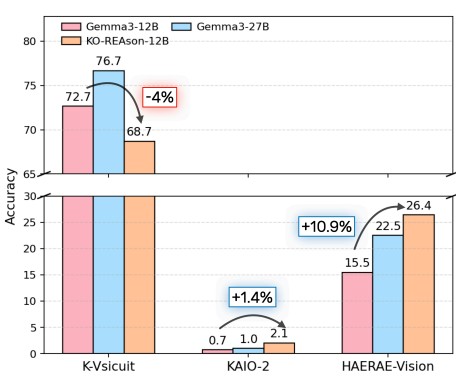

Figure 8: **Accuracy on K-Viscuit, KAIO-2, and HAERAE-Vision for Gemma3-12B, Gemma3-27B, and KO-REASON-12B.** KO-REASON-12B is a post-trained variant of Gemma3-12B on YI-SANG-HQ.

### D.3 IMPORTANCE OF HELD-IN BENCHMARKS AS PRACTICAL PROXIES.

While we apply an $n$-gram filter for decontamination, our iterative process of retraining to refine subsets inevitably uses held-in benchmarks as a proxy for progress. This raises the theoretical concern of gradually overfitting to held-in metrics. However, we view this practice as a necessary and near-optimal compromise: without a reliable proxy, it would not be possible to guide dataset construction effectively. Importantly, we do not advocate abandoning the distinction between held-in and held-out splits; both remain essential for fair evaluation. In practice (Figure 9), we find that performance gains are indeed larger on held-in benchmarks. Still, it should also be noted that gains are smaller at higher baselines overall, and part of the difference reflects the greater difficulty of held-out benchmarks. Crucially, Table 4 and Table 5 show that models trained on YI-SANG-HQ consistently improve across all benchmarks, including unseen held-out targets. This confirms that, despite mild contamination risk, our procedure achieves generalization while ensuring stable progress during training.

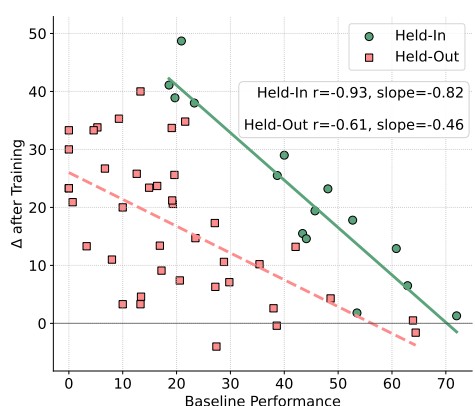

Figure 9: **Comparison of gains in Held-In/Out benchmark suites.** Each point is a (model, benchmark) pair; x-axis shows the baseline score (%), y-axis shows the improvement after training on the YI-SANG dataset. Green circles are Held-In benchmarks; red squares are Held-Out benchmarks (others). Solid/dashed lines are OLS fits.

