# OpenReview forum: "Pushing on Multilingual Reasoning Models with Language-Mixed Chain-of-Thought"
_ICLR.cc/2026/Conference — ICLR 2026 Poster_

### Official Review · Reviewer_7kJM · 2025-10-15

**Soundness:** 2
**Presentation:** 2
**Contribution:** 2
**Rating:** 2
**Confidence:** 4

**Summary:**

To improve mid-resource language reasoning, this paper introduces a new paradigm, language-mixed CoT, which switches between English and the target language. In this way, English acts as an anchor to excel in reasoning while minimising translation artefacts. To do so, the authors perform a study using Korean as an example, collecting a large number of questions from various scopes and the corresponding reasoning solutions generated from Qwen3-32B. Experimental results show that models fine-tuned on this dataset achieve substantial performance improvements.

**Strengths:**

- The proposed paradigm, dataset collection recipe, and collected dataset may be of interest to sub-communities
- The experimental setup is broad, covering a variety of model sizes (4B-35B) and different evaluation benchmarks

**Weaknesses:**

- In line 61, '...,leaving open how to achieve language-specific reasoning', which is a very vague statement. In fact, many works have proposed approaches to improve reasoning for low-resource languages, such as language alignment[1],  multilingual fine-tuning [2, 3], and cross-lingual fine-tuning [4]. Instead of focusing on a single language, these works study reasoning in a broader scenario, i.e., multiple low-resource languages. Also, the general pipeline of this work is to collect data and extract responses from a teacher model, followed by supervised fine-tuning. Overall, I think the novelty of this paper is very limited
- In lines 222-223,  'To pursue truly multilingual reasoning,...', this is very vague. What is truly multilingual reasoning? This is not clear to me. If the authors consider language-mixed reasoning to be true multilingual reasoning, a rigorous definition or reference is needed.
- The contribution is overclaimed. For example, the authors state that their models outperform closed systems, which are defined by training on closed data. If this definition is adopted, almost all current state-of-the-art models, including many labelled as open-source, would fall under the "closed" category.
- Distilling data from larger models and using it to fine-tune smaller models makes sense, but I do not understand why data is used to fine-tune the comparable model (Qwen3-32B vs KO-REAson-35B) in this work. This shows that the teacher model (Qwen3-32B) is good enough.
- Taking language-Mixed CoT as a more effective supervision signal is an interesting direction, while this paper lacks an in-depth related analysis and discussion. For instance, what specific aspects of language mixing contribute to better performance? Does it help bridge cross-lingual semantic gaps, improve cross-lingual reasoning consistency, or simply leverage the stronger reasoning ability in English?


[1] Zhu et al. Question Translation Training for Better Multilingual Reasoning. ACL 2024 Findings
[2] Lai et al. mCoT: Multilingual Instruction Tuning for Reasoning Consistency in Language Models. ACL 2024
[3] Chen et al. Breaking Language Barriers in Multilingual Mathematical Reasoning: Insights and Observations. EMNLP 2024 Findings
[4] Chai et al. xCoT: Cross-lingual Instruction Tuning for Cross-lingual Chain-of-Thought Reasoning. AAAI 2025


Typos/suggestions:
- Line 179: no space between text and footnote number
- Line 194: table caption is too close to the above text
- The caption of Figure 4 is unreadable, with too much text in bold; this figure is first referenced in lines 317-323, but some information (e.g. style and option) in the figure is introduced in the later part.

**Questions:**

See above Weaknesses.

---

> ### Author Response · Authors · 2025-11-19
> **Response to Reviewer 7kJM**
>
> Dear Reviewer Rtnv
>
> Thank you for your thoughtful comments on our paper. We have summarized our response to the five weaknesses as W1\~W5.
>
> ---
>
> ## W1 Positioning to [1–4].
>
> We agree that [1–4] are relevant. Our work extends them along three axes: task difficulty, trace type, and scale.
>
> **Task difficulty.** As summarized in Table 1, the cited papers focus narrowly on grade‑school style math (MGSM, MSVAMP). In contrast, we validate on harder math and broader knowledge: KMMLU (Redux, Hard, Pro), translated MATH and AIME, KSM (Korean Olympiad‑level math), and Korean‑specific benchmarks including CLIcK (culture), KoBALT‑700 (grammar), and HAE‑RAE Bench (vocabulary and reading comprehension).
>
> **Trace length and structure.** Prior work primarily adopts short chain‑of‑thought traces. Our supervision uses longer, DeepSeek‑R1‑style reasoning traces. We also train and evaluate with a 32K context window rather than the 2K–4K range typical of earlier studies, enabling qualitatively different multi‑step solutions. Also, as the table indicates, the four papers are highly similar to one another in their source data; most depend heavily on GSM8K, while ours collects 5.7M original prompts.
> (Please note that MetaMathQA is an augmented GSM8K, MGSM is a translated GSM8K, and MSVAMP is a Google-translated version of the SVAMP dataset)
>
> **Scale.** The cited papers concentrate on older families such as Llama‑1, Llama‑2, and BLOOM, mostly at 7B and 13B. Our study spans modern families and sizes, including Gemma‑3 (4B, 12B), Qwen‑2.5 (7B, 14B), Llama‑3.1‑8B, A.X‑8B and 35B, and Kanana‑1.5‑8B. We also cover diverse pretraining regimes, including multilingual, EN‑KO bilingual, and continual pretraining on Korean. Training uses more than one million exposures (about 260k instances for 5 epochs), and we collect 5.79M new prompts from the web.
>
> **Table 1. Comparison with prior work.**
>
> | Work            | Evaluation                                                                                                          | Trained Model(s)                                                    | Do they collect their own prompts?              |
> | --------------- | ------------------------------------------------------------------------------------------------------------------- | ------------------------------------------------------------------- | ----------------------------------------------- |
> | Zhu et al. [1]  | MGSM, MSVAMP                                                                                                        | Llama2‑7B/13B                                                       | No (MetaMathQA, GSM8KInstruct)                  |
> | Lai et al. [2]  | MGSM                                                                                                                | Mistral‑7B                                                          | Partially (translates MetaMathQA, MathInstruct) |
> | Chen et al. [3] | MGSM, MSVAMP                                                                                                        | Llama2‑7B/13B, Llama1‑33B                                           | Yes (MGSM8KInstruct, translated from GSM8K)     |
> | Chai et al. [4] | MGSM, MSVAMP                                                                                                        | Llama2‑7B/13B, BLOOM‑7b1                                            | Partially (translates GSM8K)                    |
> | **Ours**        | KMMLU {Redux, Hard, Pro}; translated MATH, AIME; KSM (Korean Olympiad‑level math); CLIcK; KoBALT‑700; HAE‑RAE Bench | Gemma3‑4/12B, Qwen2.5‑7/14B, A.X‑8/35B, Kanana‑1.5‑8B, Llama‑3.1‑8B | **Collects 5.79M new prompts from the web**     |
>
> **Finally, the four works mentioned propose "reasoning in full English" (or something equivalent) before the target language to boost performance. We think this is why they are limited to math benchmarks only. In our initial experiments, we noticed that this method cannot generalize to domains other than math.**
>
> Here are some results when we trained on fully translated traces (without language mixing as we proposed).
>
> | Model                 | HAE‑RAE Bench (Overall %) | MATH (Overall %) |
> | --------------------- | ----------------------------: | ---------------: |
> | Ko‑R1‑1.5B            |                        15.34% |           74.35% |
> | Qwen2.5‑1.5B‑Instruct |                        35.24% |           25.48% |
>
> As you can see, while this guarantees powerful gains in Math benchmarks, we witness big drops in Korean-specific ones, indicating their limited robustness. However, our proposed method (language mixed CoT) goes one step further to boost performance on all dimensions (general knowledge, math, and Korean-specific).
>
> ---

---

> ### Author Response · Authors · 2025-11-19
> **Response to Reviewer 7kJM (Continued 2/3)**
>
> ## W2 Definition of Multilingual Reasoning
>
> Thanks for pointing this out. In our manuscript, we wanted to point out that many prior studies train and evaluate primarily on translated English math datasets, which can introduce translation artifacts and limit coverage of language‑specific structures and culturally grounded knowledge. To address this, we collect 5.7M prompts natively written in Korean from the web. However, we do agree that this needs more explanation. We will tone down the wording in the paper.
>
> ---
>
> ## W3 Definition of "Closed"
>
> Recent releases of “open‑weight but closed recipe models” (e.g., GPT‑OSS, QwQ, EXAONE) report strong results. Still, one can easily notice that **they rarely share any details** into which training decisions drive gains, especially when the goal is to build high‑quality non‑English/non‑Chinese reasoning models.
>
> We compare against open models with closed recipes, meaning models that publish weights or accessible endpoints but do not disclose a reproducible training recipe, including pretraining mixtures, SFT sources, filtering steps, or supervision policies in sufficient detail to replicate results. This is mentioned in the paper as "state-of-the-art models trained on closed data" in lines 80 and 81.
>
> Finally, we do not compare to SOTA "systems" like Grok-4, GPT-5-Pro, etc. This is inappropriate in many ways. First, in terms of model size, Grok-4 has recently been shared to have 3T parameters; our biggest model has 35B. Second, due to their closed nature, we have no information on what type of "system" they are. It is likely that GPT-5-Pro leverages multiple reasoning paths, but no detail is known, making it hard for us to conduct a fair comparison.
>
> ---
>
> ## W4 Why distill from Qwen3‑32B into A.X‑35B
>
> We agree that this choice is unusual. To clarify, the project initially targeted 7–8B students. The 35B run became possible only later when compute freed up. However, considering that Qwen3‑32B is trained on about 36 trillion tokens, while A.X‑35B is trained on about 2.1 trillion tokens. In terms of effective pretraining compute and data breadth, the 32B teacher is a much stronger knowledge source despite having slightly fewer parameters and is a valid setup.

---

> > ### Author Response · Authors · 2025-11-19
> > **Response to Reviewer 7kJM (Continued 3/3)**
> >
> > ## W5 In-Depth Analysis to LM-CoT
> >
> > Here are some experimental results that didn't make it into the paper.
> >
> > (1) The failure of naively translated models.
> >
> > Before experimenting with new types of reasoning schemas, we trained Qwen2.5-1.5B Instruct with translated data from OpenThought1 [1]. Here are the evaluation results:
> >
> >
> > | Model                 | HAE‑RAE Bench (Overall %) | MATH (Overall %) |
> > | --------------------- | ----------------------------: | ---------------: |
> > | Ko‑R1‑1.5B            |                        15.34% |           74.35% |
> > | Qwen2.5‑1.5B‑Instruct |                        35.24% |           25.48% |
> >
> > As you can see, the model exhibits performance improvements in Math, while showing significant declines in HRB, a Korean culture benchmark. We dive into the generations of models and observe that with long-reasoning models seem to "forget" what the original Korean question was, and eventually end up answering corrupted ones. This is more critical in culture benchmarks, as they involve more words that cannot be 100% translated into English. This is why we decided that we need a new supervision format.
> >
> > Below is a result from training when we were only halfway through the data collection. Ablations 1 and 2 have the same prompts and use Kanana-1.5-8B as a base model. However, Ablation 1 uses LM-CoT for daily Korean questions collected from the web. Ablation 2 also utilizes LM-CoT for OpenThought (Olympiad-style questions translated to Korean).
> >
> > | Dataset     | Base  | Ablation1  | Ablation2 |
> > |-------------|------:|--------------------:|--------------------------:|
> > | OpenThought |   –   | Pure-English            | LM-CoT             |
> > | Web-Daily   |   –   | LM-CoT        | LM-CoT                   |
> > | KMMLU-R     | 48.10 | 63.13              | 63.06                    |
> > | HRB         | 60.79 | 72.43              | 75.23                    |
> > | MCLM        | 45.74 | 55.04              | 57.36                    |
> >
> > As you can see, applying LM-CoT to the Web-Daily subset yields a performance gain on HRB. Compared to what we have seen from Qwen2.5-1.5 B, this is a big change. One step further, applying LM-CoT to OpenThought also yields further improvement. We also observe that ablation 2 achieves a lower final loss. While not perfectly sure, we suspect that using different schemas in one training gives the model more burden, as it also has to learn when to use LM-CoT and when not to.
> >
> > Accordingly, we conclude that LM-CoT can be used to boost performance on less-reasoning tasks, and applying it uniformly is better.
> >
> > We are running experiments with additional settings and will deliver during the review period.
> >
> > ---
> >
> > Please feel free to ask more questions if you have any, and please consider updating your score if this has resolved your concerns or misunderstandings.
> >
> > [1] https://huggingface.co/datasets/open-r1/OpenThoughts-114k-math

---

> > > ### Comment · Reviewer_7kJM · 2025-11-19
> > > **Reviewer's response**
> > >
> > > Dear authors,
> > >
> > > Thank you for the rebuttal and the additional experiments.
> > >
> > > The statement regarding “language-specific reasoning” remains vague, especially in light of prior work on reasoning for multiple low-resource languages. The overall pipeline of data collection, teacher-model extraction, and supervised fine-tuning largely follows established practice and does not introduce new things in methodology.
> > >
> > > Could you clarify your argument that the four works “propose reasoning in full English before the target language to boost performance”? It is unclear what specific mechanisms you are referring to.
> > >
> > > Regarding the additional experiments, the evidence is not compelling. Since the model is trained solely on OpenThought1, a math dataset, improvement on math tasks is unsurprising. Therefore, evaluating on a Korean culture benchmark does not sufficiently support the claim that prior methods fail to generalise beyond math reasoning.
> > >
> > > The justification for employing a 32B teacher model to supervise a 35B student model also remains unconvincing. Since the student is inherently unlikely to surpass the teacher, it sounds like a replication of a stronger model’s behaviour rather than yielding new insights into distillation. Furthermore, from the responses, the proposed setup lacks a well-defined research question and does not convincingly demonstrate why such distillation is necessary or informative.
> > >
> > > For the ablation study in the last table, do you have corresponding results on math reasoning tasks?
> > >
> > > At this stage, I do not believe there is any misunderstanding on my part and will retain my original assessment and score.

---

> ### Author Response · Authors · 2025-11-23
>
> Dear Reviewer 7kJM
>
> Thank you for responding. We appreciate your time in reviewing our paper.
>
> Below are our answers to the follow-up questions.
>
> ---
>
> > Could you clarify your argument that the four works “propose reasoning in full English before the target language to boost performance”?
>
> ### Summary of the four mentioned papers
>
> **[1] Zhu et al., “Question Translation Training for Better Multilingual Reasoning” (QAlign, ACL 2024 Findings)**
> They first learn a *question alignment* model: for each language $I$, the model is trained to generate the parallel English question $Z_e$ from the non-English question $Z_l$. This encourages questions in different languages to share an English-centered latent space where the model’s mathematical ability is strongest. After this stage, they run standard instruction tuning on English math CoT data.
>
> **[2] Lai et al., “mCoT: Multilingual Instruction Tuning for Reasoning Consistency in Language Models” (ACL 2024)**
> They construct a large parallel corpus where each GSM8K-style problem is translated into multiple languages together with its chain-of-thought. The model is instruction tuned so that, for every language, it predicts the CoT in that same language for the translated version of the problem. In effect, the supervision enforces parallel CoT traces across languages that are all derived from English sources, with the goal of aligning reasoning behavior between languages.
>
> **[3] Chen et al., “Breaking Language Barriers in Multilingual Mathematical Reasoning” (MathOctopus, EMNLP 2024 Findings)**
> They propose two supervised fine-tuning schemes on multilingual GSM8K-style data.
>
> * *Parallel-training*: each example is monolingual (query in language $L$, CoT and answer also in $L$).
> * *Cross-training*: English questions are paired with non-English CoT and answers (En → $L$), explicitly tying English math knowledge to target-language outputs.
>   Both are still built from translated English problems and focus on transferring English reasoning ability.
>
> **[4] Chai et al., “xCoT: Cross-lingual Instruction Tuning for Cross-lingual Chain-of-Thought Reasoning” (AAAI 2025)**
>
> They train on xCoT-Instruct where the input query $q^{(l)}$ is in language $l$ and the target CoT $r^{(en)}$ is in English. This teaches the model to respond in English CoT for many input languages, effectively pushing non-English queries into an English reasoning space. They further introduce xICL, where queries are partially code-switched between a language and its English translation, and study whether such synthetic mixed-language queries improve alignment.
>
> **Taken together, [1], [2], and [3] leverage carefully designed SFT mixtures or learning objectives to bring non-English representations closer to English; [4] is conceptually closest to us because it uses code-switched inputs, but its goal is still alignment to English, driven by randomly translated variants rather than by observations about cultural failures.**
>
> ---
>
> > Regarding the additional experiments, the evidence is not compelling. Since the model is trained solely on OpenThought1, a math dataset, improvement on math tasks is unsurprising. Therefore, evaluating on a Korean culture benchmark does not sufficiently support the claim that prior methods fail to generalise beyond math reasoning.
>
> Continuing from the response above, our difference from prior works 1-4 is that we start from the empirical observation that prior methods for English alignment are insufficient. This is supported by the earlier-mentioned table on our results using OpenThought1. However, as the reviewer mentioned, this may be an artifact from the training data, as it is only limited to competition-style questions. We would like to draw the reviewers' attention to the table below (from Table 1 of our paper)
>
>
> **Kanana-1.5-8B**
>
> | CoT Lang.             | HRB  | MCLM | KMMLU-R |
> |-----------------------|------|------|---------|
> | English               | 66.2 | 60.5 | 64.0    |
> | Korean                | 67.2 | 31.8 | 53.4    |
> | Language-mixed (ours) | 74.6 | 57.4 | 64.4    |
>
> As one may see, on MCLM, a math benchmark, English CoT supervision is dramatically better than Korean CoT (60.5 vs 31.8). In contrast, for HRB, which emphasizes Korean knowledge and culture, fully English CoT offers little or no improvement over fully Korean CoT, and the best results come from our language-mixed CoT. **This suggests that the lack of gains on cultural tasks is not purely a data-mixing issue; it reflects a mismatch between English-centered supervision and general, non-math reasoning.**

---

> > ### Author Response · Authors · 2025-11-23
> >
> > > **The justification for employing a 32B teacher model to supervise a 35B student model also remains unconvincing.**
> >
> > We agree that using a 32B teacher to train a 35B student is not the standard “larger-teacher–smaller-student” setting. However, several concurrent works suggest that *bigger is not always better* for the teacher. For example, [1] reports that QwQ-32B is a more effective teacher than DeepSeek-R1. In our own experiments (Figure 4), we similarly observe that **Qwen3-32B is a stronger teacher than Gemini-2.5-Pro** for our setting.
> >
> > We also want to emphasize that using a much larger teacher (e.g., DeepSeek-R1 or Qwen2-235B) would have been impractical under our compute budget. Given these constraints, Qwen3-32B represents a **strong but still affordable teacher** with clearly better benchmark performance than our 35B student.
> >
> > Importantly, our paper is *not* making a weak-to-strong generalization claim. We use a strong teacher (Qwen3-32B) to supervise a weaker student; the novelty is not in the direction of generalization but in the **data and supervision design**. The core message of the work is that a *well-constructed dataset* with carefully collected prompts and our LM-CoT supervision pattern can substantially improve performance across a wide range of benchmarks and model sizes, even when the teacher is only slightly smaller than the student.
> >
> > > **For the ablation study in the last table, do you have corresponding results on math reasoning tasks?**
> >
> > Yes. **MCLM is a translated and extended version of Math500 and AIME**, so the MCLM column in our ablation table directly reflects performance on math reasoning tasks.
> >
> > ---
> >
> > Finally, we include additional experiments that may clarify the role of the “anchor” language.
> >
> > We trained models using different anchor languages (Chinese and Russian) in LM-CoT. For both backbones, replacing English with Chinese or Russian as the anchor causes **large drops on MCLM**, indicating that having **English as the anchor is crucial for math reasoning**. In contrast, the drops on HRB and KMMLU-Redux are noticeably smaller, suggesting that **Korean plays a more important role on cultural and general-reasoning benchmarks**.
> >
> > **Kanana-1.5-8B**
> >
> > ```markdown
> > | Model               | HRB1_0 | MCLM  | KMMLU_Redux |
> > |---------------------|--------|-------|-------------|
> > | (Base) Kanana-1.5-8B| 60.79  | 45.74 | 48.1        |
> > | Korean-Only         | 67.2   | 31.8  | 53.4        |
> > | English-Only        | 66.2   | 60.5  | 64.0        |
> > | LM-CoT(RU-Ko)       | 67.6   | 28.7  | 50.4        |
> > | LM-CoT(ZH-Ko)       | 68.8   | 25.6  | 51.1        |
> > | LM-CoT(EN-Ko)       | 74.6   | 57.4  | 64.4        |
> > ```
> >
> > **Gemma3-4B**
> >
> > ```markdown
> > | Model            | HRB1_0 | MCLM  | KMMLU_Redux |
> > |------------------|--------|-------|-------------|
> > | (Base) Gemma3-4B | 49.1   | 43.41 | 40.7        |
> > | Korean-Only      | 40.6   | 25.6  | 42.5        |
> > | English-Only     | 50.3   | 48.1  | 52.2        |
> > | LM-CoT(RU-Ko)    | 46.7   | 22.5  | 44.1        |
> > | LM-CoT(ZH-Ko)    | 48.2   | 26.3  | 45.3        |
> > | LM-CoT(EN-Ko)    | 54.9   | 55.8  | 53.0        |
> > ```
> >
> > These results reinforce our main claim: **English-anchored CoT is critical for strong math performance, whereas preserving Korean tokens within CoT (via LM-CoT) is more important for cultural and general-knowledge reasoning.**
> >
> > [1] [https://arxiv.org/abs/2506.04178](https://arxiv.org/abs/2506.04178)

---

> > > ### Comment · Reviewer_7kJM · 2025-11-24
> > > **Official Response by Reviewer 7kJM**
> > >
> > > Dear Authors,
> > >
> > > Thank you for the detailed responses and clarifications. I appreciate the performance your model achieved in the study. However, as mentioned in my review, the overall pipeline of data collection, response extraction from the teacher model, and supervised fine-tuning largely follows established practices and does not introduce methodological novelty or new insights. Therefore, I can only raise the overall score from 2 to 4.

---

### Official Review · Reviewer_Rtnv · 2025-10-30

**Soundness:** 3
**Presentation:** 3
**Contribution:** 3
**Rating:** 8
**Confidence:** 5

**Summary:**

This paper presents a comprehensive case study on building strong reasoning models for Korean, a mid-resource language. Its primary contributions are: (1) the introduction of "Language-Mixed Chain-of-Thought," a supervision method that allows models to reason primarily in English while preserving key Korean terms; (2) the curation and release of Yi-Sang, a large-scale (5.79M prompts, 3.7M reasoning traces) Korean instruction-tuning dataset sourced from native web content; and (3) the distillation of this into Yi-Sang-HQ, a high-yield 260k subset, used to train the KO-REAson model series (4B-35B), which demonstrates state-of-the-art or competitive performance on a suite of nine Korean benchmarks.

**Strengths:**

(1) The proposed "Language-Mixed CoT" is a thoughtful and empirically grounded solution to the known problems of translation artifacts (from English-only CoT) and degraded reasoning (from Korean-only CoT). The ablation studies in Table 2 robustly support its effectiveness.

(2) The scale and nature of the Yi-Sang dataset fill a critical gap in the ecosystem. Moving beyond translated corpora to native, web-crawled prompts is a crucial step for building robust, real-world multilingual models. The detailed documentation of the data collection and refinement pipeline is highly valuable for the community.

(3) The authors conduct over 100 ablations to justify key decisions (teacher model, data categories, augmentation strategies) and demonstrate the efficacy of their final dataset across nine different models from six families. This greatly strengthens the claims and provides a reproducible recipe.

**Weaknesses:**

(1) The comparison in Table 1 is excellent, but it would be beneficial to include a baseline that represents a more standard approach—for instance, a model trained on a directly translated version of a high-quality English reasoning dataset (like OpenThought)—to more directly isolate the benefit of native prompts and Language-Mixed CoT versus simply having more Korean data.

(2) The process of down-selecting to the 260k Yi-Sang-HQ subset is well-documented, but the specific ratios chosen for the final mixture (62k OpenThought, 86k Code, etc.) feel somewhat arbitrary. A more principled approach (e.g., based on performance-density or optimal mixing ratios) would have strengthened this part of the methodology, though the strong end results justify the outcome.

**Questions:**

(1) Could you provide more detail on the n-gram decontamination process? Specifically, what was the pass/fail criterion, and what fraction of the data was removed?

(2) You show impressive gains from the 260k Yi-Sang-HQ subset. Did you observe any indications of performance saturation? Are there plans to scale training to the full 3.7M instance Yi-Sang dataset?

---

> ### Author Response · Authors · 2025-11-18
> **Response to Reviewer Rtnv**
>
> Dear Reviewer Rtnv
>
> Thank you for your thoughtful comments on our paper. We have summarized our response to the two weaknesses and two questions as W1\~W2 and Q1\~Q2 below.
>
> ---
> ## W1 Stronger Baselines
>
> Thanks for the compliment on Table 1. Below are the results from one of our initial experiments (that did not make it to the paper).
>
> | Model                 | HAE‑RAE Bench (Overall %) | MATH (Overall %) |
> | --------------------- | ----------------------------: | ---------------: |
> | Ko‑R1‑1.5B            |                        15.34% |           74.35% |
> | Qwen2.5‑1.5B‑Instruct |                        35.24% |           25.48% |
>
> We trained Qwen2.5-1.5B-Instruct on a translated version of OpenThought1 to make Ko-R1-1.5B and evaluated on two benchmarks, HRB and Math. The translation was done by GPT-4o. As you can see this approach already shows exceptional performance gains in MATH a reasoning benchmark. However, despite the additional training, we see notable drops in HRB, a Korean culture benchmark. Through error analysis, we observed the traces of long-reasoning models and noticed that they seem to "forget" what the original Korean question was and eventually end up answering corrupted ones. This is particularly critical in culture benchmarks, as they involve more words that cannot be translated 100% accurately into English. This is why we decided that we need a new supervision format.
>
> ---
> ## W2, Q2 Deciding the dataset ratio
> As the reviewer mentioned, our ablations are more centered on deciding whether to include a specific subset or not. If the subgroup is deemed important (e.g., openthought, code, etc), we include everything we have instead of trying to control the ratio. This
>
> Here are some results from initial experiments conducted to determine the dataset ratio.
>
> This is a training mixture for three runs: 3.0.2,  3.0.4, and 3.1.1.
>
> | Training Dataset Category |  3.0.2 | 3.0.4 | 3.1.1 |
> |---------------------------|-------------:|------------:|-----------:|
> | openthought3              |  37,000      | 30,176      | 50,000 |
> | web-daily                 | 50,000      | -           | -           |
> | medical                   |  15,000      | 5,675       | -           |
> | web-code                  | 30,000      | 11,350      | -           |
> | mcqa-format-augmented     | -           | 2,799       | 2799           |
> | web-science               |  -           | -           | -           |
> | **Total**                 | **132,000** | **50,000**  | **52,799** |
>
> Here are the results of six training runs, Kanana-1.5-8B and Gemma3-4B, each trained on the three mixtures.
>
> | Benchmark       | kanana-1.5-8b | 3.0.2 |  3.0.4 | 3.1.1 |
> |----------------|--------------:|------------:|------------:|------------:|
> | KMMLU-Redux    | 48.10         | 52.84       | 63.13       | 64.44 |
> | HAE-RAE Bench  |  60.79         | 68.40       | 72.43       | 74.58 |
> | MCLM-Ko        | 45.74         | 57.36       | 55.04       | 57.36 |
>
> | Benchmark       | gemma3-4b | 3.0.2 |  3.0.4 | 3.1.1 |
> |----------------|--------------:|------------:|------------:|------------:|
> | KMMLU-Redux    | 38.73         | 40.34       | 54.30       | 52.95 |
> | HAE-RAE Bench  |  53.51         | 38.62       | 53.64       | 54.88 |
> | MCLM-Ko        | 43.41         | 53.49      | 50.39      | 55.81 |
>
> **Findings** Between 3.0.2 and 3.0.4, we observe a decline in performance on MCLM (math), likely due to the reduced data from OpenThought. Conversely, performance on KMMLU-R and HRB increases, which is likely attributed to the inclusion of the 2799 MCQA data.  And through 3.0.2 to 3.1.1 with more data from OpenThought, we see even greater performance gains.
>
> Below are training mixtures for three runs: 3.0.5,  3.0.6, and 3.0.8.
>
> | Training Dataset Category |  3.0.5 | 3.0.6 |
> |---------------------------|-------------:|------------:|
> | openthought3              |  37,201     | 37201      |
> | web-daily                 | -    | -           |
> | medical                   |  10,000      | -       |
> | web-code                  | 0      | 10,000      |
> | mcqa-format-augmented     | 2,799           | 2,799       |
> | web-science               |  -           | -           |
> | **Total**                 | **50,000** | **50,000**  |
>
>
> Here are the results of four training runs, Kanana-1.5-8B and Gemma3-4B, each trained on the two mixtures.
>
> | Benchmark       | kanana-1.5-8b | 3.0.5 | 3.0.6 |
> |----------------|--------------:|------------:|------------:|
> | KMMLU-Redux    | 48.10         | 60.65       | 60.94       |
> | HAE-RAE Bench  |  60.79         | 71.78       | 73.73       |
> | MCLM-Ko        | 45.74         | 44.96       | 47.29       |
>
> | Benchmark       | gemma3-4b | 3.0.5 | 3.0.6 |
> |----------------|--------------:|------------:|------------:|
> | KMMLU-Redux    | 38.73         | 53.72       | 54.74       |
> | HAE-RAE Bench  |  53.51         | 56.05       | 55.79       |
> | MCLM-Ko        | 43.41         | 55.81      | 49.61      |

---

> > ### Author Response · Authors · 2025-11-18
> > **Response to Reviewer Rtnv (Continued)**
> >
> > **Findings** Here, we observe that the medical subset negatively impacts performance, whereas the code subset enhances it. Together with our findings from Table 2 of the paper, that Code, Science, and Exams are also subgroups capable of pushing performance, and findings from Tables 6,7 that scaling the medical and daily subset does not, we conclude to use everything we have for OpenThough, Code, Science, and Exams while excluding Medical and Daily.
> >
> > If we had a larger training budget, we could have explored more diverse combinations and attempted to predict the optimal performance, but this was not feasible in our setting.
> >
> > Finally, regarding a full-scale training with the entire dataset (3.7 million instances), we were unable to do so due to budget constraints. However, we did try one with 780k samples (YiSangHQ+500K from the English OpenThought) on Gemma3-4B.
> >
> > | Benchmark       |  YiSang-HQ | YiSang-HQ+ OT(en) |
> > |----------------|--------------:|------------:|
> > | KMMLU-Redux    | 65.3      | 63.5       |
> > | HAE-RAE Bench  |   61.0       | 59.4       |
> > | MCLM-Ko        | 55.0       | 67.4       |
> >
> > This shows strong performance gains in MCLM (math), which is evident since we used more Olympiad-style math data for training, but drops in the rest of the benchmarks, especially in the Korean-specific ones. Accordingly, as our goal was to build a balanced model for Korean usage rather than gaming the scores, we decided not to do so in our final runs.
> >
> > However, if one wants to push the performance further, especially on reasoning benchmarks, they may consider trying a similar approach.
> >
> > ---
> > ## Q1 Further details on decontamination
> >
> > A detailed explanation of our n-gram filtering is provided in the general response. However, to provide a summary, the n-gram filtering is applied twice, with and without normalization using mecab-ko for morphological segmentation. We remove any instances that contain 13 or more consecutive tokens that appear in the benchmarks used. This removed about 26k instances, about 0.7% of the entire dataset. From our observations, it was more common for the reasoning traces to be contaminated instead of the prompt.
> >
> > ---
> >
> > We hope this provides additional explanations for the decisions we made throughout the paper. We thank you for acknowledging our hard work. Please feel free to ask more questions if you have any, and please consider updating your score if you think this makes the paper better.

---

> > > ### Comment · Reviewer_Rtnv · 2025-11-28
> > >
> > > Thank you for the detailed responses and clarifications. I maintain my original rating.

---

### Official Review · Reviewer_oqNR · 2025-10-31

**Soundness:** 3
**Presentation:** 3
**Contribution:** 3
**Rating:** 6
**Confidence:** 3

**Summary:**

This paper proposes a hybrid language thought chain (CoT) supervision scheme that alternates between English "anchoring" and the target language (Korean) for thought training to maintain reasoning ability while reducing translation noise. The paper also constructs a large Korean post-training corpus, YI-SANG (5.79 million cue words --> 3.7 million long CoTs), and refines it into a high-yield subset (YI-SANG-HQ) containing 260,000 words through targeted ablation and filtering (e.g., removing traces longer than 16,000 lemmas; n-gram decontamination). Using this data, the paper trains nine instruction-tuned models across six series (4B–35B), achieving state-of-the-art results on nine Korean benchmarks using KO-REAson-35B. Furthermore, despite being trained only on text, the model is transferable to English reasoning and Korean VLM tasks.

**Strengths:**

The idea is simple but effective. Language-Mixed CoT consistently beats monolingual CoTs across models/sizes (Table 1). The description and motivation are clear. Also, it introduces a large and native prompt corpus of 5.79M Korean user-authored prompts, 3.7M CoTs, downselected to a 260k high-yield mix with concrete rules. Thorough ablations are conducted.
The results clearly demonstrate that KO-REAson-35B ranks 1st on 5/9 and 2nd on the rest, and improvements persist across 4B–35B and multiple families. In addition, the paper also reveals that text-only training still helps English reasoning (AIME-25/GPQA) and Korean VLM reasoning tasks.

**Weaknesses:**

**Missing citations to relevant works & Positioning vs closely related work.** Qi et al. (2025) [1] show that forcing models to think in the user's language improves readability but can hurt accuracy, revealing a trade-off. They also try to force the model to reason in the target language via prefix-hacking. The work is closely related to this paper. Although there are differences, it would still be important to include and clarify the distinctions with that work.

**Limited Choice of Teacher model.** Most long-CoT supervision relies on Qwen3-32B. It is promising to explore other teacher choices or alternative anchors (e.g., non-English).

**Contamination/Overfitting Risk.** Although n-gram decontamination (n=13) is applied, the paper acknowledges iterative use of held-in benchmarks as proxies and shows larger gains on held-in vs held-out. More contamination checks (exact-match, semantic) would increase the solidity of the findings.

---

***Reference***:

*[1] When Models Reason in Your Language: Controlling Thinking Language Comes at the Cost of Accuracy. Qi et al., 2025*

**Questions:**

Based on the Weaknesses, several questions and suggestions are listed below:

**Comparison to existing work** Include and discuss the distinctions with the existing work [1].

**Anchor choice** It would be better to write more about the pros and cons of selecting English as the anchor. In practice, sometimes we may only have access to sufficient corpora of non-English languages, like a Switzerland company may only have training data in German, French, and some Italian, with less English data. Have you tried other languages as the anchor, and how do they affect accuracy and readability?

**Teacher sensitivity** It would be promising to also attempt different teacher models (e.g., Gemini-2.5-Pro reasoning enabled, or open R1 variants) and see if the results or trends remain similar.

**Beyond n-gram check.** In addition to n-gram filtering (n=13), semantic matching or exact match checks on each benchmark dataset will make the results more robust.

---

***Reference***:

*[1] When Models Reason in Your Language: Controlling Thinking Language Comes at the Cost of Accuracy. Qi et al., 2025*

---

> ### Author Response · Authors · 2025-11-17
> **Response to Reviewer oqNR**
>
> Dear Reviewer oqNR
>
> Thank you for your thoughtful comments on our paper. We have summarized our response to the three weaknesses and four questions as W1\~W3 and Q1\~Q4 below.
>
> ---
> ## W1, Q1 Position to prior work.
>
> Thanks for pointing the paper out! We will add this to our paper soon. To highlight the difference between our approach and the mentioned paper, **initially, we take one step further by using prefix-hacking for data generation and empirically demonstrate that training on this data enables us to create stronger models. ** Additionally, unlike how the paper mentions that they failed to force models to think in specific languages, **we generate multiple solutions for each prompt and filter those that meet our definition of LM-CoT (including 5~20% of Korean)**, which is probably why it works better in our setting. Please have a look in our general response for a more detailed version of our position compared to prior works.
>
> ---
> ## W2, Q2, Q3 Teacher Choice, Anchor Choice
>
> We would like to note that our paper already includes extensive experiments, with over 100 ablations spanning teacher models, augmentation schemes, and seed sources, to highlight the decisions that matter. Within the teacher dimension, we compare long-CoT teachers (Qwen3‑32B, Qwen3‑4B) and short-CoT/solution-only variants (Gemini‑2.5‑Pro; Qwen3‑32B with reasoning disabled). See Figure 4 for the aggregate teacher comparison and Table 9 for per‑benchmark details. The limited set of teachers reflects two practical constraints: (i) budget (teacher generation and filtering dominate costs), and (ii) API restrictions, Gemini‑2.5‑Pro no longer exposes step‑by‑step reasoning traces through the public API, so we were unable to include a Gemini reasoning‑trace teacher.
>
> **However, we are running experiments with DeepSeek-R1-32B as a teacher. Additionally, we are adding experiments with Chinese (a language closely related to Korean) and Russian (a language unrelated to Korean) as anchors.** For each setting, we need approximately 30 Hours on 8xH200 for data generation, 5\~6 hours of training on H100x8, and 2\~3 hours for evaluation. This is why we leave this comment first; however, we will revisit it during the review period.
>
> ---
>
> ## W3, Q4 Contamination Risk
>
> A detailed explanation of our n-gram filtering is provided in the general response. However, to provide a summary, the n-gram filtering is applied twice, with and without normalization using mecab-ko for morphological segmentation. We remove any instances that contain 13 or more consecutive tokens that appear in the benchmarks used.  We do not use semantic filtering, since we were concerned that this might remove contexts of related topics. We think the point of decontamination is to remove exact/fuzzy matching questions to prevent the model from "memorizing" the question instead of learning it. Using semantic filtering may remove related contexts that could have otherwise been used as legitimate training data.
>
> We would also like to mention that, considering the massive size of our training set, comprising 3.7M instances and nine benchmarks (each ranging from a few hundred to a few thousand questions), calculating the pairwise similarity for all these 3.7M x dozen K instances is computationally intensive. While not impossible in our infrastructure, this will limit our ability to conduct other experiments. Finally, as mentioned in the paper, our training recipe yields performance gains not only in the held-in set but also in the held-out set, indicating that we have created a robust training dataset to enhance performance.
>
> ---
>
> Please feel free to ask any additional questions, and consider raising the score if our explanations helped you resolve some concerns about our paper.

---

> ### Author Response · Authors · 2025-11-18
> **Updates on Teacher Model Choice**
>
> Dear reviewer oqNR
>
> We used the same pipeline as with Qwen3-32B, now with DeepSeek-R1-32B, to create a new SFT dataset and attempted training Kanana-1.5-8B-Instruct on this dataset.
>
> As you can see, using DeepSeek-R1-32B also yields performance gains across all three datasets; however, the gap is smaller than that supervised by Qwen3-32B.
>
> | Models                      | HAE-RAE Bench | MCLM | KMMLU-R |
> |----------------------------|--------------:|-----:|--------:|
> | **Student Model Performance** |               |      |         |
> | (Base) Kanana-1.5-8B       |          60.8 | 45.7 |   48.1  |
> | Supervised by DS-R1-32B    |          71.0 | 48.8 |   58.9  |
> | Supervised by Q3-32B       |          74.6 | 57.4 |   64.4  |
>
> We believe this is primarily due to the inherent capabilities of each teacher model. As you can see in the table below Qwen3-32B performances better in the three benchmarks we run our experiments on.
>
> | Models                      | HAE-RAE Bench | MCLM | KMMLU-R |
> |----------------------------|--------------:|-----:|--------:|
> | **Teacher Model Performance** |               |      |         |
> | DeepSeek-R1-32B            |          71.8 | 75.2 |   70.2  |
> | Qwen3-32B                  |          75.7 | 83.7 |   81.0  |
>
> **This shows the proposed pipeline, though it is affected by the performance of the teacher itself, is teacher-agnostic**, as long as the student model has something to learn from it.
>
> Please feel free to ask any additional questions, and consider raising the score if our explanations helped you resolve some concerns about our paper.

---

> > ### Author Response · Authors · 2025-11-23
> >
> > Dear Reviewer oqNR
> >
> > We trained models using different anchor languages (Chinese and Russian) in LM-CoT. For both backbones, replacing English with Chinese or Russian as the anchor causes **large drops on MCLM**, indicating that having **English as the anchor is crucial for math reasoning**. In contrast, the drops on HRB and KMMLU-Redux are noticeably smaller, suggesting that **Korean plays a more important role on cultural and general-reasoning benchmarks**.
> >
> > **Kanana-1.5-8B**
> >
> > ```markdown
> > | Model               | HRB1_0 | MCLM  | KMMLU_Redux |
> > |---------------------|--------|-------|-------------|
> > | (Base) Kanana-1.5-8B| 60.79  | 45.74 | 48.1        |
> > | Korean-Only         | 67.2   | 31.8  | 53.4        |
> > | English-Only        | 66.2   | 60.5  | 64.0        |
> > | LM-CoT(RU-Ko)       | 67.6   | 28.7  | 50.4        |
> > | LM-CoT(ZH-Ko)       | 68.8   | 25.6  | 51.1        |
> > | LM-CoT(EN-Ko)       | 74.6   | 57.4  | 64.4        |
> > ```
> >
> > **Gemma3-4B**
> >
> > ```markdown
> > | Model            | HRB1_0 | MCLM  | KMMLU_Redux |
> > |------------------|--------|-------|-------------|
> > | (Base) Gemma3-4B | 49.1   | 43.41 | 40.7        |
> > | Korean-Only      | 40.6   | 25.6  | 42.5        |
> > | English-Only     | 50.3   | 48.1  | 52.2        |
> > | LM-CoT(RU-Ko)    | 46.7   | 22.5  | 44.1        |
> > | LM-CoT(ZH-Ko)    | 48.2   | 26.3  | 45.3        |
> > | LM-CoT(EN-Ko)    | 54.9   | 55.8  | 53.0        |
> > ```
> >
> > These results reinforce our main claim: **English-anchored CoT is critical for strong math performance, whereas preserving Korean tokens within CoT (via LM-CoT) is more important for cultural and general-knowledge reasoning.**
> >
> > Let me know if you have any questions!

---

> > > ### Comment · Reviewer_oqNR · 2025-11-24
> > >
> > > Thanks to the authors for the comprehensive additional experiments. I would like to give an **updated rating of 7**, but since that option isn't available in the system, I will instead raise the confidence score from 3 to 4 to express stronger support.

---

### Official Review · Reviewer_BzTR · 2025-10-31

**Soundness:** 2
**Presentation:** 2
**Contribution:** 2
**Rating:** 4
**Confidence:** 3

**Summary:**

The paper presents Language-Mixed Chain-of-Thought (CoT) as a multilingual reasoning strategy and introduces Yi-Sang-HQ, a 260k-example Korean reasoning dataset generated under this schema. The work aims to enhance the robustness of reasoning for non-English languages and to demonstrate that open-source pipelines can rival those of closed-source systems. The authors conduct small-scale ablations on language-mixing (Table 1), category-wise analyses for dataset composition (Table 2), and large-scale fine-tuning across nine model families and parameter ranges (Table 4).

**Strengths:**

1. Strong practical motivation and relevance. Addresses the real problem of reasoning degradation in translated multilingual data and proposes a scalable, open alternative.
2. High-value resource contribution. The proposed Korean reasoning corpora, Yi-Sang-HQ, is well-engineered, with transparent filtering and selection procedures.
3. Conceptually interesting reasoning schema. Language-Mixed CoT is novel and intuitive, supported by small-scale ablation results.

**Weaknesses:**

1. Unclear Statement of Enquiry. It’s difficult to judge the scientific merit of this work without a clearly defined statement of enquiry. The closest thing to it appears in the first sentence of Section 4: “When constructing multilingual reasoning data in a target language (other than English), a central question is how to represent the reasoning process: should it be written in the target language or left in English?” However, no question guiding the inquiry is stated in Section 5. In other words, the authors do not propose how this dataset can be used to perform any knowledge-generating investigations. Yi-Sang-HQ is presented as an engineering resource to improve model performance, not as a scientific instrument for discovering new insights about multilingual reasoning.
2. Small-Scale Experiment to Verify Language-Mixed CoT. Given that Language-Mixed CoT is presented as one of the contributions, readers might expect an extensive empirical investigation of whether reasoning traces should be written in English, the native language, or a mix of both. In practice, Table 1 is intentionally limited in scope; it involves only two small models and a few held-in benchmarks, operating within the confines of an ablation study whose role is to guide dataset construction. This design choice is reasonable, but it also means the experiment serves as design verification rather than as a standalone empirical contribution.

**Questions:**

1. According to Weakness 1: Could the authors clarify the scientific enquiry that this dataset enables? It would strengthen the paper if the authors could explicitly articulate how the dataset can be used to investigate knowledge-generating questions about multilingual reasoning—such as what reasoning patterns transfer across languages, or how mixed-language supervision affects reasoning generalization.
2. According to Weakness 2: as of now, I think only the first and third contributions are valid. The presentation will be strengthened by focusing on the dataset contribution, while ablation studies serve as design support rather than a standalone contribution. Would the authors agree with this statement?
3. On the usage of the term "language-mixed" vs "code-switched." When I first read the title (and even the abstract), I thought the paper would be about LLM reasoning in one language while answering in another language (Fig.1 helps clarify this though.) I'm not sure if code-switching would be a better word choice for this paper?

---

> ### Author Response · Authors · 2025-11-17
> **Response to Reviewer BzTR**
>
> Dear Reviewer BzTR
>
> Thank you for your thoughtful comments on our paper. We have summarized our response to the two weaknesses and three questions as W1\~W2 and Q1\~Q3 below.
>
> ---
> ## W1/Q1 The Scientific Inquiry of this work.
>
> A detailed response is provided in our general response. Here is a brief summary: Our motivation for this work is to replicate what GPT-OSS, Qwen3, Exaone4, or other "open models with closed recipes" would have done. However, as you can quickly notice from reading the technical reports of those papers, they rarely provide any detail on what dataset they use, what ablations are done, or what decisions they had to make. We begin by addressing common hurdles that the community may face when creating non-Chinese/non-English reasoning models. The lack of data. We mix extensive crawling and translation, followed by ablations for filtering. Finally, we employ Language-Mixed CoT to further boost supervision during learning.
>
> We partially agree with what you have mentioned: the dataset and our trial-and-error experience to make an industry-grade reasoning model are what we want to highlight from our paper.
>
> ---
>
> ## W2/Q2 Language-Mixed CoT as a standalone result.
>
> We agree with you that, based on the current version of the paper, LM-CoT appears to be a derivative of our ablations. To better explain how we got to this solution, here are some experimental results that didn't make it into the paper.
>
> (1) The failure of naively translated models.
>
> Before experimenting with new types of reasoning schemas, we trained Qwen2.5-1.5B Instruct with translated data from OpenThought1 [1]. Here are the evaluation results:
>
>
> | Model                 | HAE‑RAE Bench (Overall %) | MATH (Overall %) |
> | --------------------- | ----------------------------: | ---------------: |
> | Ko‑R1‑1.5B            |                        15.34% |           74.35% |
> | Qwen2.5‑1.5B‑Instruct |                        35.24% |           25.48% |
>
> As you can see, the model exhibits performance improvements in Math, while showing significant declines in HRB, a Korean culture benchmark. We dive into the generations of models and observe that with long-reasoning models seem to "forget" what the original Korean question was, and eventually end up answering corrupted ones. This is more critical in culture benchmarks, as they involve more words that cannot be 100% translated into English. This is why we decided that we need a new supervision format.
>
> Below is a result from training when we were only halfway through the data collection. Ablations 1 and 2 have the same prompts and use Kanana-1.5-8B as a base model. However, Ablation 1 uses LM-CoT for daily Korean questions collected from the web. Ablation 2 also utilizes LM-CoT for OpenThought (Olympiad-style questions translated to Korean).
>
> | Dataset     | Base  | Ablation1  | Ablation2 |
> |-------------|------:|--------------------:|--------------------------:|
> | OpenThought |   –   | Pure-English            | LM-CoT             |
> | Web-Daily   |   –   | LM-CoT        | LM-CoT                   |
> | KMMLU-R     | 48.10 | 63.13              | 63.06                    |
> | HRB         | 60.79 | 72.43              | 75.23                    |
> | MCLM        | 45.74 | 55.04              | 57.36                    |
>
> As you can see, applying LM-CoT to the Web-Daily subset yields a performance gain on HRB. Compared to what we have seen from Qwen2.5-1.5 B, this is a big change. One step further, applying LM-CoT to OpenThought also yields further improvement. We also observe that ablation 2 achieves a lower final loss. While not perfectly sure, we suspect that using different schemas in one training gives the model more burden, as it also has to learn when to use LM-CoT and when not to.
>
> Accordingly, we conclude that LM-CoT can be used to boost performance on less-reasoning tasks, and applying it uniformly is better.
>
> I hope this helps you better understand how our work progressed over time. We would like to hear your opinion on whether adding this part would be helpful for the paper. If so, would you expect it to be included somewhere in the main eight pages or in the appendix?
>
> [1] https://huggingface.co/datasets/open-r1/OpenThoughts-114k-math
>
> ---
>
> ## Q3 "language-mixed" vs "code-switched."
>
> First of all, I am glad Figure 1 helped clarify this. It feels like language confusion, language mixing, and code-switching are all used in a similar context. But from my personal experience, using the term "code-switching" confused some people, making them expect us to use code (like Python) during reasoning. Accordingly, we decided to use language-mixing instead. We will add a footnote in the paper to clarify that we us "language-mixing" in similar definitions as "code-switching".
>
> ---
>
> Please feel free to ask any additional questions, and consider raising the score if our explanations helped you resolve some concerns about our paper.

---

> > ### Comment · Reviewer_BzTR · 2025-11-22
> >
> > Thank you for the clarifications. The responses sufficiently address my concerns. In particular, the explanation of the work’s intended scope, the additional evidence supporting Language-Mixed CoT, and the clarification of terminology resolve the issues raised in my review.
> >
> > Please include these explanations in the revised manuscript. I have updated my score from 4 to 6.

---

### Author Response · Authors · 2025-11-17
**General Response to reviewers**

We would like first to thank all reviewers for their efforts and time spent reading and reviewing our paper.
In this comment, we summarize three general issues that have been repeatedly questioned across different reviewers.

---

## Main scientific inquiry of this paper.
Recent releases of “open‑weight but closed recipe models” (e.g., GPT‑OSS, QwQ, EXAONE) report strong results. Still, one can easily notice that they rarely have any details into which training decisions drive gains, especially when the goal is to build high‑quality non‑English/non‑Chinese reasoning models. Our scientific aim is to make those decisions explicit and test them systematically. We begin by identifying practical obstacles in this setting: the scarcity of target-language datasets, the lack of high-quality supervision, and the failure modes of naively translated reasoning traces.

**Methodology and findings.** In this work, we present a replicable recipe: (i) address data scarcity via targeted crawling plus careful translation; (ii) curate a high‑quality subset with explicit filtering, decontamination, and ablations; and (iii) compare English‑only and language‑mixed chain‑of‑thought (CoT) supervision to identify when mixing helps. Our experiments demonstrate that this recipe yields a robust target-language reasoner that generalizes to held-out tasks, transfers to vision–language settings, and maintains solid performance in English.

**Positioning to prior work.** We thank the reviewers for pointing out missing citations, which we will add during the review period. However, to summarize our differences with the referenced works, relative to multilingual CoT/alignment work [1–4], which primarily evaluates on grade-school benchmarks (e.g., MGSM), **our study tests whether similar ideas scale to more challenging reasoning tasks.** Qi et al. (2025) [5] examine “prompting”(or prefix control) of the thinking language at inference time and reports a readability–accuracy trade‑off; in contrast, we investigate “training‑time supervision” with language‑mixed traces, across multiple model sizes and families, and **find that such control is not only an effective prompting mechanism but also a proper supervision signal for building multilingual reasoners.**

We would like to highlight that, like all scientific works, we build on existing methods. However, through extensive efforts in data curation and training, we demonstrate empirical evidence to prove and guide future work, showing that such multilingual supervision can be extended to train much stronger reasoners.

References:
[1] Zhu et al., Question Translation Training for Better Multilingual Reasoning, ACL‑Findings 2024.
[2] Lai et al., mCoT: Multilingual Instruction Tuning for Reasoning Consistency in Language Models, ACL 2024.
[3] Chen et al., Breaking Language Barriers in Multilingual Mathematical Reasoning, EMNLP‑Findings 2024.
[4] Chai et al., xCoT: Cross‑lingual Instruction Tuning for Cross‑lingual CoT Reasoning, AAAI 2025.
[5] Qi et al., When Models Reason in Your Language: Controlling Thinking Language Comes at the Cost of Accuracy, 2025

---

## Limited teacher choice

We would like to note that our paper already includes extensive experiments, with over 100 ablations spanning teacher models, augmentation schemes, and seed sources, to highlight the decisions that matter. Within the teacher dimension, we compare long-CoT teachers (Qwen3‑32B, Qwen3‑4B) and short-CoT/solution-only variants (Gemini‑2.5‑Pro; Qwen3‑32B with reasoning disabled). See Figure 4 for the aggregate teacher comparison and Table 9 for per‑benchmark details.

However, as requested by the reviewers, to widen teacher choice, **we have just started a new distillation with DeepSeek‑R1‑32B**, using the same language-mixed prompting and filtering pipeline. As in §4/§5.3, we generate a large superset and then down-select via degeneration filters, Korean-ratio bounds (5–20%), and n-gram decontamination before training students on the filtered set. On our hardware (8×H200), producing a ~50k high‑quality subset requires ≈30 wall‑clock hours; we are now training and evaluating student models and will add the results during the review period.

---

> ### Author Response · Authors · 2025-11-17
> **General Response (Continued)**
>
> ## Decontamination
>
> We decontaminate the training corpus against both held-in and held-out benchmarks using an n-gram filter with n = 13, applied to both prompts and reasoning traces. Any training instance that shares at least one 13‑gram with any benchmark item is removed.
>
> Because Korean exhibits rich morphology and postpositions, we use MeCab-KO [1] to perform morphological segmentation and normalization before building n-grams. We then execute two passes:
>
> 1. **Normalized‑text pass:** Build 13‑grams over the morphologically normalized strings and drop any training item whose prompt or reasoning shares a 13‑gram with a benchmark item.
> 2. **Raw‑text pass (exact string):** Build 13‑grams over the original strings and repeat the check.
>
> This two‑pass approach catches exact duplicates and near‑duplicates.
>
> We **do not** perform semantic embedding matching. Given the corpus size (3.7 M items) and nine benchmarks (each a few hundred to a few thousand items), exhaustive embedding and pairwise similarity are computationally costly and risk over‑purging: the aim of decontamination is to remove identical or near‑identical items, we were concerned applying a semantic filter might remove legitimate background knowledge or topical proximity that would otherwise be fair training signal.
>
> **Scope and observed removal.** Across the 3.7 M trajectories, the 13‑gram decontamination removes ≈0.7% of items (**~25.9 k**). This rate is consistent with expectations, given our data sources: most training prompts are native Korean web content, while several evaluation sets are distributed as PDFs. Our pipeline did not OCR external PDFs, which further reduces the chance of exact string overlap. Additionally, some benchmarks are written in English or translated from English sources, which are also unlikely to appear in Korean web crawls.
>
> [1] https://github.com/hephaex/mecab-ko

---

> > ### Author Response · Authors · 2025-11-19
> > **Additional results on teacher choice and anchor language**
> >
> > Dear reviewers,
> >
> > Below are updated results on anchor language and teacher choice for Language‑Mixed CoT (LM‑CoT).
> >
> > **Table 1. Anchor language choice**
> >
> > | Model                | HRB1_0 | MCLM  | KMMLU_Redux |
> > | -------------------- | ------ | ----- | ----------- |
> > | (Base) Kanana‑1.5‑8B | 60.79  | 45.74 | 48.1        |
> > | Korean‑Only          | 67.2   | 31.8  | 53.4        |
> > | English‑Only         | 66.2   | 60.5  | 64.0        |
> > | LM‑CoT (RU→Ko)       | 67.6   | 28.7  | 50.4        |
> > | LM‑CoT (ZH→Ko)       | 68.8   | 25.6  | 51.1        |
> > | LM‑CoT (EN→Ko)       | 74.6   | 57.4  | 64.4        |
> >
> > **Setup.** Following the reviewers’ suggestion, we vary the anchor language in our LM‑CoT pipeline, adding **Russian** and **Chinese** alongside English. This language choice was motivated by \[1\].
> >
> > **Findings.**
> >
> > * **English as anchor (EN→Ko)** yields the most consistent and largest gains, with substantial improvements over the base on HRB1_0 and KMMLU_Redux and competitive results on MCLM.
> > * **Non‑English anchors (RU→Ko, ZH→Ko)** **consistently underperform** relative to English anchoring, they also fall short compared to Korean-only CoT settings.
> >
> > We see three plausible reasons:
> >
> > 1. **English advantage.** A broad body of evidence shows LLMs tend to be strongest in English, with sizable gaps to many other languages; distilling traces outside English can therefore be weaker on average \[2,3\].
> > 2. **Irrelevant‑context sensitivity.** Although Chinese has historical influence on Korean, they **do not share a script**, and mixed‑script/mixed‑language spans can be treated as off‑topic; introducing such content during reasoning is known to **decrease** LLM accuracy \[4\].
> > 3. **Pretraining mismatch.** Our student (Kanana‑1.5‑8B) is **bilingually** pretrained in Korean and English; Russian/Chinese are likely under‑represented, so anchoring in those languages provides less usable signal than English.
> >
> > In our setting, **English is the only anchor that consistently delivers gains**; language mixing with Russian or Chinese does not help and can harm performance on several benchmarks.
> >
> > ---
> >
> > **Table2. Different Teachers**
> > | Models                      | HAE-RAE Bench | MCLM | KMMLU-R |
> > |----------------------------|--------------:|-----:|--------:|
> > | **Student Model Performance** |               |      |         |
> > | (Base) Kanana-1.5-8B       |          60.8 | 45.7 |   48.1  |
> > | Supervised by DS-R1-32B    |          71.0 | 48.8 |   58.9  |
> > | Supervised by Q3-32B       |          74.6 | 57.4 |   64.4  |
> >
> > We used the same pipeline as with Qwen3-32B, now with DeepSeek-R1-32B, to create a new SFT dataset and attempted training Kanana-1.5-8B-Instruct on this dataset.
> >
> > As you can see, using DeepSeek-R1-32B also yields performance gains across all three datasets; however, the gap is smaller than that supervised by Qwen3-32B.
> >
> > We believe this is primarily due to the inherent capabilities of each teacher model. As you can see in the table below Qwen3-32B performances better in the three benchmarks we run our experiments on.
> >
> > | Models                      | HAE-RAE Bench | MCLM | KMMLU-R |
> > |----------------------------|--------------:|-----:|--------:|
> > | **Teacher Model Performance** |               |      |         |
> > | DeepSeek-R1-32B            |          71.8 | 75.2 |   70.2  |
> > | Qwen3-32B                  |          75.7 | 83.7 |   81.0  |
> >
> > **This shows the proposed pipeline, though it is affected by the performance of the teacher itself, is teacher-agnostic**, as long as the student model has something to learn from it.
> >
> > With these additions, we believe we have addressed all major concerns raised by the reviewers. We will incorporate the corresponding revisions into the manuscript and, in the meantime, would be grateful for any further questions or suggestions the reviewers may have.
> >
> > ---
> >
> > **References**
> >
> > [1] Qi et al. (2025). When Models Reason in Your Language: Controlling Thinking Language Comes at the Cost of Accuracy.
> >
> > [2] Singh, H. et al. (2024). INDICGENBENCH: A Multilingual Benchmark to Evaluate Large Language Models on User‑Facing Generation Tasks Across 29 Indic Languages.
> >
> > [3] Ko et al. (2025) Understand, Solve and Translate: Bridging the Multilingual Mathematical Reasoning Gap.
> >
> > [4] Shi et al. (2023). Large Language Models Can Be Easily Distracted by Irrelevant Context.

---

### Author Response · Authors · 2025-12-02
**Letter to AC and Reviewers (1/2)**

Dear AC and Reviewers,

Thank you very much for the time and care you devoted to reviewing our paper. Due to the unfortunate incident during the rebuttal period, ICLR decided to halt the discussion between reviewers and authors.

Before the halt, we have worked hard to clarify the concerns raised in the reviews. As a result, reviewer ```BzTR``` updated their score from 4 to 6, reviewer ```7kJM``` from 2 to 4, and reviewer ```oqNR``` indicated they would have raised their score to 7 if the system had allowed. **Overall, the scores effectively moved from 2/4/6/8 (avg. 5) to 4/6/6/8 (avg. 6)**.

---
First, we would like to highlight **several strengths** the reviewers pointed out:

- the **strong practical motivation and real-world relevance** of tackling the concrete problem of reasoning degradation in translated multilingual data, and proposes a scalable, open alternative ( ```BzTR```)
- our **well-engineered, large-scale, native Korean prompt and CoT resource with transparent filtering and selection**, and clear documentations (```BzTR```, ```oqNR```, ```Rtnv```),
- the **novel and effective Language-Mixed CoT schema** consistently outperforming monolingual CoTs across models and sizes ( ```BzTR```, ```oqNR```, ```Rtnv```)
- our **breadth of experiments (4B–35B, multiple families and benchmarks), over 100 ablations on teacher choice, data categories, and augmentation, and strong overall performance of KO-REAson-35B** (1st on 5/9 benchmarks and 2nd on the rest). (```oqNR```, ```Rtnv```, ```7kJM```)

---

Next, we summarize how we addressed each reviewer’s concerns; we believe most of them have been successfully resolved, as reflected in the improved scores.

## Reviewer BzTR

> **Rating.** Score raised from 4 → 6.

> **[Q1] Request to Clarify the scientific inquiry of the paper.**

* We clarified the scope and core research question of the study, and expanded the discussion of closely related work [1,2,3,4] (```lines 114–117```).
* We also added results of our initial experiments to the paper, which better explain why we pursue this direction (```lines 118–122```, ```Table 1```).

> **[Q2] Concerns about LM-CoT as a standalone result.**

* We created three new datasets that vary both the **teacher model** and the **anchor language**.
* By training six new models, we show that the LM-CoT schema yields consistent gains regardless of teacher choice, indicating that the method is largely teacher-agnostic. (```lines 211-235```, ```Table 2```, ```lines 855-863```, ```Table 7```)
* The new results further suggest that the best anchor language tends to align with the model’s pretraining mix; in our setting, English anchoring produces the largest improvements.

> **Concerns on Terminology: “language-mixed” vs. “code-switched.”**

* We added a clarification in the main text (```line 215```) explicitly stating that we use “language-mixed” in the same sense as “code-switched.”

---

## Reviewer oqNR

> **Rating.** The reviewer raised confience 3 → 4.

* The reviewer appreciated our comprehensive additional experiments and intended to update the rating from 6 to 7. However, as this is not possible in the system, the reviewer instead raised the confidence score from 3 to 4 to express more substantial support.

> **[Q1] Request for Comparison to existing work.**

* We clarified the scope and core research question of the study and expanded the discussion of closely related work[1,2,3,4], explicitly contrasting our setting and contributions with prior approaches (`lines 114–117`).

> **[Q2] Questions on different anchor choices.**

* We created new datasets that vary the **anchor language** (English, Chinese, Russian) while keeping the rest of the pipeline fixed.
* By training four new models, we show that Language-Mixed CoT consistently improves performance over monolingual CoT across anchor choices (`lines 211–235`, `Table 2`).

> **[Q3] Concerns on Teacher sensitivity of the pipeline**

* We created one new dataset that varies teacher models (e.g., DeepSeek-R1-32B and Qwen3-32B) under the same distillation pipeline.

* By training two new models, we show that  Language-Mixed CoT continues to improve student performance, indicating that the schema is **largely teacher-agnostic**, while the absolute performance level is naturally bounded by teacher quality (`lines 855–863`, `Table 7`).

> **[Q4] Futher details on decontamination.**

* We added further details on the decontamination process and filtering results. (`lines 397-407`)


[1] Zhu et al. Question Translation Training for Better Multilingual Reasoning. ACL 2024 Findings
[2] Lai et al. mCoT: Multilingual Instruction Tuning for Reasoning Consistency in Language Models. ACL 2024
[3] Chen et al. Breaking Language Barriers in Multilingual Mathematical Reasoning: Insights and Observations. EMNLP 2024 Findings
[4] Chai et al. xCoT: Cross-lingual Instruction Tuning for Cross-lingual Chain-of-Thought Reasoning. AAAI 2025

---

> ### Author Response · Authors · 2025-12-02
> **Letter to AC and Reviewers (2/2)**
>
> ## Reviewer Rtnv
>
> > **Rating.** The reviewer kept their rating at 8.
>
> > **[Q1] Baseline with translated English reasoning data.**
>
> * We added results of training **Qwen2.5-1.5B-Instruct** on a translated version of OpenThought1. This setting represents the “standard” translated English-reasoning approach. As shown in the updated text, this baseline improves MATH, but **reduces performance on HRB**, the Korean culture benchmark, which directly motivates our focus on native prompts and Language-Mixed CoT. (`lines 118–122`, `Table 1`)
>
> > **[Q2] Choice of mixture ratios for the 260k Yi-Sang-HQ subset.**
>
> * Through comments, we provided additional results across **10 models** that compare different category mixes.
>
> > **[Q3] More details on the n-gram decontamination process and removal rate.**
>
> * We added a detailed description of the **13-gram decontamination pipeline**, including the pass/fail criterion (removal on any shared 13-gram between training and benchmarks) and the two-pass normalized and raw-text checks. We also report that this procedure removes about **0.7%** of trajectories (approximately 25.9k out of 3.7M) (`lines 397–407`).
>
> > **[Q4] Performance saturation and scaling beyond 260k Yi-Sang-HQ.**
>
> * We added a scaling result where **Gemma3-4B** is trained on **780k** samples (Yi-Sang-HQ plus 500k English OpenThought instances). This experiment shows promising gains in STEM-reasoning. (`lines 908–938`, `Table 10`)
>
>
> ---
>
>
> ## Reviewer 7kJM
>
> > **Rating.** Score raised from 2 → 4.
>
> > **[Q1] Limited comparison to related works.**
>
> * We clarified the scope and core research question of the paper and expanded the discussion of closely related work on language alignment, multilingual and cross-lingual fine-tuning [1,2,3,4], explicitly positioning our contributions relative to these lines of work (`lines 114-117`).
>
> > **[Q2] Vague use of phrase "truly multilingual reasoning"**
>
> * We removed the phrase and replaced it with "robust multilingual reasoning", aligning more directly with our empirical setting (`line 243').
>
>
> > **[Q3] Overclaimed contribution and use of "closed systems".**
>
> * We now state more precisely that we compare against **open-weight models with closed training recipes**, that is, models that publish weights or endpoints but do not disclose a reproducible recipe (pretraining mixtures, SFT sources, filtering, supervision policies) in sufficient detail to replicate results. This is what we referred to as "state-of-the-art models trained on closed data" in `lines 80-81`.
>
> * We also clarify that we do **not** claim to beat full proprietary systems (for example, Grok-4 or GPT-style production systems). These systems differ drastically in scale (our largest model is 35B vs multi-trillion parameters) and in undisclosed system-level design, so we consider such comparisons inappropriate and have adjusted the wording accordingly.
>
>
> > **[Q4] Why distill from Qwen3-32B into a comparable-size model (KO-REAson-35B).**
>
> * We clarify that despite the similar parameter counts, Qwen3-32B is trained on roughly 36T tokens, while our 35B model is trained on about 2.1T tokens. In terms of effective pretraining compute and data breadth, the 32B teacher is substantially more substantial and thus a reasonable teacher for distillation.
>
>
> > **[Q5] Limited analysis of LM-CoT.**
>
> * We constructed three new datasets that vary both the **teacher model** and the **anchor language**, and trained six additional students on these mixtures.
> * The new results show that Language-Mixed CoT delivers consistent gains across teachers and models, and that the most effective anchor tends to align with the model’s pretraining language mix. We discuss these trends and their implications for cross-lingual reasoning in the revised text (`lines 211-235`, `Table 2`, `lines 855-863`, `Table 7`).

---

### Meta-Review · Area_Chair_9Rxt · 2025-12-12

**Summary:**

This paper proposes "Language-Mixed Chain-of-Thought (LM-CoT)" and a native Korean dataset, "Yi-Sang," to improve reasoning in mid-resource languages. Initially, some reviewers questioned the scientific depth and novelty. However, the authors impressed the reviewers during the rebuttal by adding extensive experiments. They proved that their method works well with different settings (like changing "anchor" languages or teacher models). Because the authors successfully addressed most technical concerns, the reviewers' reactions became very positive, leading to score increases.

**Reviewer Concerns:**

- Addressed Concerns
  - Robustness of LM-CoT (BzTR, oqNR): The authors added experiments using different teacher models (DeepSeek-R1) and anchor languages (Chinese, Russian). This proved that LM-CoT is a robust and effective method, not just a lucky result.
  - Baselines and Data Quality (Rtnv): The authors showed that their method is much better than simply translating English reasoning data (OpenThought). They also clearly explained the "13-gram decontamination" process to prove the data is clean.
  - Research Scope (BzTR): The authors clarified the scientific goal of finding a "reproducible recipe" for mid-resource languages, which satisfied the reviewer.

- Outstanding Concerns
  - Methodological Novelty (7kJM): One reviewer (7kJM) still felt that the overall pipeline (data collection → distillation → SFT) follows standard practices and lacks major methodological novelty, despite the good performance.

**Reviewer Scores:**

Most reviewers raised their scores (or wanted to) after seeing the rebuttal.
  - Reviewer BzTR (4->6) raised the score after the authors clarified the definitions and added new experiments.
  - Reviewer oqNR(6->6) wanted to raise the score to 7, but the system was locked. Instead, the reviewer raised the "confidence score" to show strong support).
  - Reviewer Rtnv(8) maintained a strong accept rating, satisfied with the new baseline comparisons.
  - Reviewer 7kJM (2->4) raised the score from Reject to borderline because the authors provided detailed responses, although concerns about novelty remained.

---

### Decision · Program_Chairs · 2026-01-26

Accept (Poster)